# Sensitivity Experiments on the Response of Vb Cyclones to Sea Surface Temperature and Soil Moisture Changes

Martina Messmer[1,2], Juan José Gómez-Navarro[1,2,3], and Christoph C. Raible[1,2]

[1]Climate and Environmental Physics, Physics Institute, University of Bern, Bern, Switzerland
[2]Oeschger Centre for Climate Change Research, University of Bern, Bern, Switzerland
[3]now at Department of Physics, University of Murcia, Murcia, Spain

*Correspondence to:* Martina Messmer (messmer@climate.unibe.ch)

**Abstract.** Extra-tropical cyclones of type Vb, which develop over the western Mediterranean and move northeastward, are major natural hazards being responsible for heavy precipitation over Central Europe. To gain further understanding in the governing processes of these Vb cyclones the study explores the role of soil moisture and sea surface temperature (SST) and their contribution to the atmospheric moisture content. Thereby, recent Vb events identified in the ERA-Interim reanalysis are dynamically downscaled with the Weather Research and Forecasting model (WRF). Results indicate that a mean summer high-impact Vb event is mostly sensitive to an increase in the Mediterranean SSTs and rather insensitive to Atlantic SSTs and soil moisture changes. Hence, an increase of +5 K in Mediterranean SSTs leads to an average increase of 24 % in precipitation over Central Europe. This increase in precipitation is mainly induced by larger mean upward moisture flux over the Mediterranean with increasing Mediterranean SSTs. This further invokes an increase in latent energy release, which leads to an increase in atmospheric instability, i.e., in convective available potential energy. Both, the increased availability of atmospheric moisture and the increased instability of the atmosphere, which is able to remove extra moisture from the atmosphere due to convective processes, are responsible for the strong increase in precipitation over the entire region influenced by Vb events. Precipitation patterns further indicate that a strong increase in precipitation is found at the eastern coast of the Adriatic Sea for increased Mediterranean SSTs. This premature loss in atmospheric moisture leads to a significant decrease in atmospheric moisture transport to Central Europe and the northeastern flanks of the Alpine mountain chain. This leads to a reduction in precipitation in this high-impact region of the Vb event for an increase in Mediterranean SSTs of +5 K. Furthermore, the intensity of the Vb cyclones, measured as a gradient in the 850-hPa geopotential height field around the cyclone centre, indicates that an upper bound for intensity might be reached for the most intense Vb event.

## 1 Introduction

The frequency and intensity of extreme events are highly vulnerable to climate change (Hartmann et al., 2013; Fischer and Knutti, 2014; Fischer et al., 2014; Fischer and Knutti, 2015), e.g., heavy precipitation events in the midlatitudes exhibit an increase with on-going climate change (Hartmann et al., 2013). Since it is difficult to predict changes of extreme weather events, in particular at regional scales in a possible future climate (Fischer and Knutti, 2015), it is of great importance to

understand the triggering mechanisms and the involved processes of high-impact weather events, e.g., cyclonic systems with their associated wind gusts and heavy precipitation.

A prominent phenomenon of regional high-impact weather in Central Europe, and especially over the northern ridge of the Alps and the adjacent flatlands and low mountain ranges, is the so-called Vb cyclone. Vb events are known as cyclones that typically develop over the Mediterranean Sea (Gulf of Genoa) and travel during their intensification phase along the southern side of the Alps. As they reach the eastern edge of the Alpine mountain chain, they turn north-eastward towards St. Petersburg (Van Bebber, 1891). These cyclones transport large amounts of atmospheric moisture to the northern side of the Alps and Central Europe, thus triggering extreme precipitation events (Messmer et al., 2015) and exhibit a great potential for floods in the Elbe, Danube or also the Rhine catchment (Nied et al., 2014) and the Alpine area including adjacent flatlands and low mountain ranges (e.g., chapter 5 in MeteoSchweiz, 2006).

Several studies record that often cutoff-lows, including the Vb pathway, are responsible for extreme precipitation and discharge events in the Alps and Central Europe, e.g., the prominent European flood that occurred in August 2002 (Ulbrich et al., 2003a; Jacobeit et al., 2006; Grams et al., 2014; Messmer et al., 2015; Awan and Formayer, 2016). The potential of transporting extreme precipitation to Central Europe is especially high if these cutoff-low systems are positioned in the northern or eastern part of the Alps (Awan and Formayer, 2016). These studies above demonstrate that there seems to be a wide agreement on the large-scale dynamics of Vb events. Furthermore, the large-scale dynamics seem to determine whether a Vb cyclone delivers high precipitation or not (Messmer et al., 2015). Despite this fact, an important moisture source needs to supply the atmosphere with the required moisture. In fact, these thermodynamical processes, and especially the moisture sources, remain unclear as described in the following.

To identify the main moisture sources during Vb events, the case study approach is widely used in the literature (Ulbrich et al., 2003a; Stohl and James, 2004; Sodemann et al., 2009; Gangoiti et al., 2011). The most intensively studied Vb cyclone is the one-in-a-century event that occurred in August 2002 and led to a major flooding of the Oder and Elbe catchment. Some studies have identified evaporation from land, together with moisture from the Mediterranean Sea and the Atlantic, as important moisture sources during the 2002 Vb event (Ulbrich et al., 2003a; Stohl and James, 2004). This is in line with the study performed by Sodemann et al. (2009), who suggested that water vapour from separated moisture sources contributes to the extreme precipitation in the most affected area during the August 2002 Vb event. These moisture sources include the Atlantic Ocean and Mediterranean Sea areas inside the model domain, the evapotranspiration from land areas, and long-range advection from subtropical areas outside the model domain. However, some more general studies on precipitation events in Europe suggest that the Mediterranean Sea plays an important role in such events. Gimeno et al. (2010), for example, identified the Mediterranean Sea as the main oceanic moisture source for precipitation over Central Europe. Gangoiti et al. (2011) focused on the August 2002 Vb event and identified evaporation in the Western Mediterranean basin 6 to 2 days prior to the actual event as its most prominent source of moisture. Winschall et al. (2014) further supported the fact that the Mediterranean Sea is not the only moisture source during various heavy precipitation events in Central Europe. They found that additional moisture sources with high event-to-event variability are needed to trigger such events. These moisture sources include beside others the evaporation from European landmasses especially in summer or evaporation from the North Atlantic Ocean in winter. The fact

that evaporation from land, and thus soil moisture recycling, might play an important role in extreme precipitation events has been further highlighted in recent studies (Grams et al., 2014; Kelemen et al., 2016). Both studies analyse a rather atypical Vb event in 2013, which was nevertheless associated with widespread flooding in the Danube and Elbe catchment. Even though there have been several case studies devoted to identify the moisture sources during high-impact Vb events, the results seem to be diverse as the moisture sources include the Mediterranean Sea, the Atlantic Ocean and soil moisture. Therefore, identifying the main moisture source during Vb events in general and independent of single cases, still remains a challenge.

A one-at-a-time sensitivity experiment can help identifying the main moisture sources as it allows to diagnose the processes that contribute most to the model parametric sensitivity (Lee et al., 2012; Zhao et al., 2013). Thus, sensitivity analyses enable analysing the impact of several factors on a certain process (Saltelli et al., 2000). Consequently, the effect on, e.g., precipitation can be determined according to changes in the input variable, e.g., sea surface temperatures (SSTs).

The present work aims at shedding light on the sensitivity of extreme summer Vb events and their impact on precipitation over Central Europe to several moisture sources. Hence, a number of idealised sensitivity experiments are designed and carried out with the regional Weather Research and Forecasting Model (WRF) to disentangle the contribution of these moisture sources during the five most intense summer Vb events (Messmer et al., 2015) recorded in the period 1979–2013. Thereby, and according to the variables considered by previous studies, we test the sensitivity of Vb events to changes in soil moisture in Europe and SSTs of the Atlantic Ocean and the Mediterranean Sea.

The structure of the study is as follows. Details on the model setup, data set and applied methods are presented in Sect. 2. Section 3 provides a short evaluation of the control simulation, while the results of the sensitivity experiments are discussed in Sect. 4. In Sect. 5, we focus on the Mediterranean sensitivity experiments, including an analysis of changes in cyclone tracks and characteristics. Finally, a summary of the main conclusions and a short outlook is presented (Sect. 6).

## 2 Data and Methods

### 2.1 Reanalysis data set

The ERA-Interim data set is used to provide the initial conditions and 6-hourly lateral boundaries for the regional model. This data set is produced by the European Centre for Medium Range Weather Forecast (ECMWF) in a spectral resolution of T255, which corresponds to a spatial resolution of approximately 80 km, and 60 vertical levels up to 0.1 hPa (Dee et al., 2011). The 6-hourly estimates of three-dimensional meteorological variables and the 3-hourly estimates for surface variables are generated with the Integrated Forecast System model version 2006 of the ECMWF assimilating various sources of observational data, e.g., satellite data, surface pressure observations, and radiosonde profiles (section 4 in Dee et al., 2011).

### 2.2 Observations used in the model evaluation

For evaluation, simulated daily accumulated precipitation and multi-day sums of daily accumulated precipitation over the five precipitation-intense summer Vb events are compared to two observational data sets. The first one is the E-OBS data set version

10.0 (Haylock et al., 2008). It consists of weather station data, which are interpolated to a regular 25 km grid over the European land, i.e., it does not provide data over the ocean. The variables included in this product are: precipitation, sea level pressure, and mean, minimum and maximum temperature. All variables have daily resolution and span the period 1950–2013 (Haylock et al., 2008). For our analysis we will only use the daily accumulated precipitation.

The second data set is the EURO4m-APGD precipitation data. It contains the daily accumulated precipitation distribution over the European Alps and the adjacent flatland regions for the period 1971–2008 (Isotta et al., 2014). In contrast to E-OBS, the data is based on measurements from high-resolution rain-gauge stations and thus provides 5-km resolution on a regular grid in the ETRS89-LAEA coordinate system (Isotta et al., 2014).

## 2.3    Selection of Vb events

For this analysis, five precipitation-intense summer Vb events are selected in the period between 1979 to 2013 that triggered extreme precipitation over the region of the northern slope of the Alps and northern Central Europe. For that the ERA-Interim period between 1979 and 2013 is used to identify several Vb events by applying a tracking tool developed by Blender et al. (1997) to the geopotential height field at 850 hPa (Messmer et al., 2015). The Vb tracks are then filtered with a technique adapted from Hofstätter and Chimani (2012). The filtered Vb events are classified and sorted according to the accumulated

precipitation delivered over the region of the northern Alps, including parts of Switzerland, Austria, Germany and the Czech Republic. More details on the method of Vb event selection are presented in Messmer et al. (2015).

The selected five most precipitation-intense summer Vb cyclones include two events that are of historic importance. One event is the so-called European Flood, that happened in August 2002 and especially affected the catchment areas of two rivers: the Elbe and the Oder (Ulbrich et al., 2003a, b). The other event took place in August 2005, and caused severe floods on the

northern side of the Alps, especially in Switzerland (MeteoSchweiz, 2006). The other three events occured in July 1981, August 1985 and in June 1979. These three events are not related to historic flooding events. All of the five events are initialised by a cold air outbreak located northeast of the Alps. As this trough moves westwards lee-cyclogenesis is induced at the southeastern flanks of the Alps and hence in the region of the Gulf of Genoa. From this starting point all of the five analysed Vb cyclones move along the Vb track described by Van Bebber (1891), showing some individual behaviour along the path of course.

## 2.4    Model setup and sensitivity experiments

The simulations for the sensitivity experiments are carried out with the Weather Research and Forecasting Model (WRF), version 3.5.1. WRF is run with a three-nested domain setup with a nest ratio of 1:3. The domains have a spatial resolution of 27, 9 and 3 km and are 2-way nested, which allows feedbacks from the higher to the lower resolution domains (Fig. 1). The outermost domain covers all of the Mediterranean Sea and a large part of the Atlantic Ocean. The design of the domains considers

a large area of water masses to be included in the outermost domain in order to allow strong water vapour signals in the inner domains. Hence, although the innermost domain does not include the Atlantic, the outer domains allow WRF to consistently integrate the moisture flux provided by the physical mechanisms outside the smallest domain. This flux is advected towards Central Europe through the various domain boundaries. The innermost domain targets Central Europe, showing the Alpine

mountain chain, and thus the region of interest, in the middle of the domain (Fig. 1). Vertically, all simulations implement 50 eta levels. The 3-km resolution in the innermost domain allows the explicit simulation of convective processes, so no additional parameterisation is needed. Other important parameterisations chosen to run the WRF simulations are listed in Table 1.

Nudging techniques are avoided (except for the pre-simulations for the moisture sensitivity simulations, see details below), so that Vb cyclones can freely develop their path and intensity, according to the new boundary conditions imposed by the sensitivity experiments. However, the fact that nudging is not admitted, renders the starting time of the simulation critical, since too early initialisations may lead to situations where the Vb cyclone is very different to the one reproduced in ERA-Interim, or even completely missing. After testing several initiation times (not shown), we found that starting the simulation six hours before the corresponding event is observed first, allows reproducing the events. This means, the simulated trajectory of the cyclone mimics the corresponding track of the events found in the original ERA-Interim data set (Messmer et al., 2015). However, this relatively short spin-up period of six hours can be a drawback as the model might not be in full equilibrium. Note that the spin-up time is equal for all three domains, which means that there is no additional time lag for the nested domains.

To assure that this short spin-up period does not affect the performance of the simulation in the sensitivity studies, a set of experiments was performed with a spin-up time of one week. The set of experiments consists of sensitivity simulations where SST changes of -5 K and +5 K in the Atlantic and the Mediterranean Sea are applied (not shown). These tests are aimed to assess to what extent longer simulations can achieve a better equilibrium state, leading to different results. To force the model to reproduce the Vb event and circumvent the problem stated above, the wind fields (U and V) and the geopotential height (GPH) are spectrally nudged (wavelengths larger than roughly 600 km) above the planetary boundary layer and in domain 1 only. Note that nudging has only been applied to this one week spin-up setup. We found hardly any change in the thermodynamic variables when using this longer spin-up period (not shown). We thus conclude that the length of the spin-up period is suitable to reach an equilibrium during the whole life of the Vb event.

### 2.4.1 Sensitivity experiment for soil moisture

To test the sensitivity of Vb events to soil moisture, three different experiments are carried out. They enclose a complete desaturation to the minimum possible soil moisture content of 2 %, fixing a soil moisture content typical of southern Spain homogeneously across the whole domain (i.e. 17.5 %, which corresponds to the average value of the analysed Vb events in the region of Ciudad Real, being this one of the driest regions in the Iberian Peninsula), and a complete saturation of the soil moisture content. The second setup is a very unrealistic, yet physically plausible scenario, and therefore can be regarded as a more realistic version of the complete desaturation of soil. Note that all of the three performed experiments are rather unrealistic and highly idealised, and are aimed at exploring physical mechanisms, rather than obtaining accurate climate change projections. A complete desaturation of all the soil moisture throughout whole Europe is probably the most unrealistic one, since it comprises soil water contents over whole Europe that do not even occur in the Saharan Desert. It is possible that Central Europe could see a general drying to a more Mediterranean climate (Seneviratne et al., 2010), but nevertheless it is quite a strong reduction in soil water content for most of the land area covered by domain 1. In this sense, the second experiment is a slightly less unrealistic version, although still very unlikely given current climate change projections in the Mediterranean

Sea. The projections for Central Europe indicate a robust 5–15 % reduction in soil moisture for the end of the century, with a tendency for wetter soils in the northern parts of Europe (Seneviratne et al., 2010). This projected reduction is still higher than the southern Spain experiments, as a reduction of 15 % of the Central European soil moisture would result in our cases in 25 % soil water content. Similarly, the full saturation experiment is also rather unrealistic even in a possible moistening scenario of Europe.

Since the evaporation from soil moisture can influence the moisture content in the atmosphere before the actual Vb event takes place, we have carefully designed the initialization of these simulations. For all the three experiments, as well as the control simulation, the WRF model is started five days before the actual Vb event is initialized and terminated after these five days. During this pre-simulation, we use the same spectral nudging as described for the SST test simulations, and the soil moisture is constantly overruled to impose a fixed value of soil moisture according to each of the three sensitivity experiments in all four model layers of the Noah model. The atmospheric water vapour content after these five days of the pre-simulation is then used to overwrite the water vapour present in the initial conditions taken from the driving dataset and used in the actual Vb simulation. The actual Vb event simulations are started at the same time as the SST experiments in order to obtain similar cyclone tracks throughout the different types of experiments and therefore minimise side effects arising from changes in the Vb dynamics.

Across all soil moisture sensitivity experiments, the initial conditions for soil moisture in the actual simulation are set to the corresponding value according to each the three families described above. In this regard, it is important to note that just the initial conditions are set, i.e. the model is free to adjust the soil moisture afterwards due to e.g., precipitation and evaporation processes. For this reason, we did not use spin-up times longer than 6 hours, since otherwise the model would use the longer spin-up period to refill the soil moisture volume until the equilibrium is recovered. Further, such short spin-up precludes obtaining a realistic initial condition of the water atmosphere content in equilibrium with the perturbed soil, which is the reason for running the pre-simulations described above. The care taken in the initialization of the soil experiments pertains especially the first model soil layer, which is the most weather-relevant layer and the one with shortest response time. It is important to remark that unlike in the SST experiments, where we change a given boundary condition, in the case of soil the variables are simulated together with the atmosphere model, and therefore the soil experiments shall be regarded as perturbation in the initial conditions. To change the soil moisture content for the actual Vb event simulation, the original ERA-Interim initial file is modified and the land values are set to either 0, 0.175 or 0.5 $m^3$ $m^{-3}$. The latter value is selected, because the soil moisture content of all soil types listed in the WRF model is always lower than 0.5 $m^3$ $m^{-3}$.

The full saturation soil experiment described above, represents an averaged increase in land soil moisture by 21 % compared to the control simulation for the first soil layer, which is the most relevant for weather. In contrast, the complete drainage experimental setting and the southern Spain soil experiment reduces soil moisture by 68.5 % and 24 %, respectively, when temporally and spatially averaging domain 3.

### 2.4.2 Sensitivity experiment for the Atlantic SST

In order to gain insight into the moisture impact of the Atlantic SSTs on Vb events, the Atlantic SSTs are increased and decreased by 5 K. The two most extreme sensitivity experiments are performed to obtain a strong signal in the results. Since this large change in the Atlantic SSTs does not strongly impact precipitation (Sect. 4 for more details) other sensitivity experiments

with lower SST amplitudes are not performed.

The increase of the Atlantic SSTs in our experiment has been chosen according to the increase in SSTs in the sensitivity experiments of the Mediterranean SSTs described in Sect. 2.4.3. This is, to obtain some consistency within the two families of the SST sensitivity experiments.

### 2.4.3 Sensitivity experiments for the Mediterranean SST

For the sensitivity experiments within the Mediterranean Sea, ten sensitivity simulations plus a control simulation are performed for each of the five Vb events. This corresponds to homogeneous SST changes within the Mediterranean Sea between -5 K and +5 K, in one-degree intervals (0 K is the control simulation). The ERA-Interim SST field is used to calculate the horizontally interpolated SST field for the input file used by WRF. The homogeneous increase in SSTs is added to the horizontally interpolated WRF grid obtained after WPS and not to the original ERA-Interim data set itself. This is done to avoid any

inconsistencies in the increased Mediterranean SSTs at grid points close to the coast lines, related to differences in the land-sea mask of the ERA-Interim and WRF domain.

Compared to the reference period 1961–1990, Mimura et al. (2007, chapter 16.3) projected for the fossil intensive A1 scenario a maximal warming of the Mediterranean open ocean surface air by up to 2.19 K, 3.85 K and 7.07 K for the time periods 2010–2039, 2040–2069 and 2070–2099, respectively. Additionally, Shaltout and Omstedt (2014) expected an annual

warming of the Mediterranean Sea by 2.6 K and for the summer season a warming of 2.9 K by the end of the 21[th] century under the RCP8.5 scenario, being this a worst-case scenario that involves very pessimistic scenario emissions and leads to severe climate change projections. Hence, the warming implied in the sensitivity experiments are in line with the spread of projected scenarios for several periods of the 21[st] century.

### 3  Model evaluation of the control simulations

The control simulations of the five analysed Vb events are used as reference for the different sensitivity experiments in the following. As this analysis shall show the ability of WRF to realistically reproduce such events, key variables of these control simulations are compared to observational data sets and ERA-Interim data. The analysis focuses on precipitation and the trajectories of the Vb events.

## 3.1 Precipitation

To show the performance of WRF in simulating Vb cyclones and their impact we first focus on precipitation. Daily accumulated precipitation and multi-day sums of daily accumulated precipitation are evaluated in two different areas.

First, both variables are compared to observations for the entire domain 3 using E-OBS. For this the E-OBS data set is
bilinearly interpolated onto the grid of the innermost domain and the ocean grid points are masked, since the E-OBS data is land only. For the comparison, the simulated and observed mean daily accumulated precipitation for five Vb events are shown in Fig. 2(a). WRF generally simulates higher daily accumulated precipitation compared to E-OBS across all five days of the Vb events. These differences are mainly caused by an overestimation of the simulated precipitation during the first two days of each event, and coincide with the highest daily accumulated precipitation. As a consequence of this, the multi-day sums
of daily accumulated precipitation in domain 3 is systematically higher for WRF throughout all the selected Vb events than E-OBS in domain 3 (Fig. 2(b)). This mismatch can be attributed to some extent to deficiencies in the E-OBS data, since it is known that precipitation is underestimated in the E-OBS data, especially over mountain areas and during summer (Hofstra et al., 2009). The reason is that precipitation is mainly driven by convection during summer, and thus it is very local, making it difficult to capture these phenomena with the sparse observation network that is available over the Alps (Hofstra et al.,
2009). Additionally, some of the overestimation by WRF can be attributed to the finer resolution compared to E-OBS. Hence, lower values are expected for the coarser E-OBS grid, as each grid point represents an average over a larger area compared to the WRF grid (Göber et al., 2008). Furthermore, possible positive biases in the average precipitation of the regional model additionally increase the differences between E-OBS and WRF.

Second, the same variables are compared in a smaller area focusing over the Alps, which is depicted by the "Alps" box in
Fig. 1. In this case, the simulated extreme daily accumulated precipitation compared to E-OBS and EURO4m-APGD, tend to line up around the one-to-one relationship (second row in Fig. 2) indicating a close resemblance between the observed and simulated daily accumulated precipitation during the different Vb events. The same is also true for the multi-day sums of daily accumulated precipitation during the complete event. Note that as indicated before WRF overestimates daily accumulated precipitation compared to E-OBS, whereas it generally underestimates precipitation compared to EURO4m-APGD data
(Fig. 2(d)). This opposite behaviour of E-OBS and EURO4m-APGD compared to WRF underlines the argument about the uncertainties in the E-OBS data set as explanation for the mismatch between simulated and observed precipitation for domain 3. Indeed, the EURO4m-APGD data set includes a denser spatial network of the rain-gauge stations. This renders it more suitable to capture the local convective systems that predominantly occur during summer and that lead to the high amounts of precipitation that are simulated by WRF but are not captured by E-OBS.

The evaluation indicates that WRF is able to realistically capture the daily accumulated precipitation and thus, also the multi-day sums of daily accumulated precipitation during the five precipitation-intense summer Vb events of interest. Further, the fact that WRF overestimates precipitation compared to E-OBS underlines the ability of WRF to accurately simulate convective processes over the Alpine area.

## 3.2 Cyclone track

To evaluate the cyclone trajectories obtained by WRF, the tracks are compared to the ones observed in ERA-Interim data. The latter are detected by a tracking tool (Blender et al., 1997) applied to the 1.5°×1.5° resolved 850-hPa geopotential height field (see Messmer et al., 2015). Since the downscaled geopotential height field is affected by high frequency noise, which is introduced by the fact that the domains are located over the Alps, the track detection is applied to the outermost domain only. The 850-hPa geopotential height field is bilinearly interpolated onto a regular latitude-longitude grid with 0.5°×0.5° resolution to smooth the field and remove the high-frequency noise. Nevertheless, the resolution is still somewhat finer than the ERA-Interim grid.

The tracks of the control simulation (light green line in Fig. 3) agree well with the ones obtained by ERA-Interim (black line in Fig. 3) in all of the five analysed Vb events. In particular during the intensification phase of a cyclone, i.e., the first time steps, the alignment with the ERA-Interim tracks is obvious, even though a slight displacement towards the south is noticeable. In the decaying phase of the cyclone more deviations from the ERA-Interim path are found. Note that the precipitation intense time steps happen during the intensification phase of the cyclone and therefore a deviation from the ERA-Interim at the end of the cyclones' life time does not strongly influence the precipitation amounts, i.e., the key variable in our analysis.

## 4 Sensitivity of Vb cyclones to soil moisture, Atlantic and Mediterranean SSTs

In the following we present the analysis of the different idealised sensitivity experiments focusing on daily mean precipitation, moisture flux over land and the Mediterranean Sea, precipitable water and convective available potential energy (CAPE). These variables are able to provide insight into the processes that take place within the moisture exchange from its sources to the atmosphere. Therefore, all variables are averaged over domain 3 (tests with areas encircled around the cyclone center by $\geq$ 500 km show similar results). Since most of the ocean grid points are located over the Mediterranean Sea and only few over the Atlantic (see domain 3 in Fig. 1), these few grid points have been masked to obtain only the moisture flux over the Mediterranean Sea.

The time steps that are included in the analysis are defined by the time when 95 % of the total precipitation of the event has fallen over the "Alps" box depicted in Fig. 1. This allows studying the impact of the Vb event itself, and avoids a potential contamination of the analysis due to the development of other weather phenomena, such as frontal systems, in the decaying phase of the Vb cyclone. Domain 3 represents the influence area of the different Vb cyclones and it is therefore the region of main interest. The statistical confidence of the differences between the sensitivity experiments and the control simulations is established with the non-parametric Mann–Whitney–U test at the 5 % significance level.

### 4.1 Soil moisture

The idealised soil moisture experiment reveals that a complete drainage of the soil moisture volume in the initial conditions leads to an average reduction of 32 % in the daily mean precipitation over the five studied Vb events in the area of domain 3

(Fig. 4(a)). The sensitivity experiments corresponding to soil moisture as in Southern Spain show a small decrease of around 6 % in daily mean precipitation. In contrast, a fully saturated soil moisture volume in the initial conditions leads to a relatively small increase of 7 % with respect to the control simulation. The daily mean upward moisture flux over land decreases by approximately 81 % and 14 % for a complete drainage of the soil moisture volume and the southern Spain soil conditions, respectively. At the same time the daily mean upward moisture flux over land shows an increase of 11 % for full saturation (Fig. 4(b)). As expected, the daily mean upward moisture flux over the Mediterranean Sea, precipitable water and CAPE reveal only small changes for the two experiments with the soil moisture volume and consequently, they do not show significant changes (Fig. 4(c)–(e)). Therefore, the reduction in precipitation as well as in precipitable water with a complete drainage can be attributed to a reduction of moisture flux from the land (Fig. 4(b)), which is in turn a direct consequence of the complete removal of the soil moisture volume.

The reason is that a reduction (increase) in soil moisture volume leads to a reduction (increase) in latent heat flux and therefore to an increase (reduction) in sensible heat flux. This further decreases (increases) precipitation, since relative humidity over land is strongly modified during these experiments (not shown). There is a slight reduction (increase) in the mean upward moisture flux over the Mediterranean Sea. These changes are not significant and hence, their changes are not analysed in more detail here.

The average spatial precipitation patterns obtained within the soil experiment show a strong reduction in the continental precipitation for the complete drainage experiment compared to the control simulation (Fig. 5(a)). Especially higher elevated regions are affected by the decrease in precipitation such as the Alpine mountain ridge or the Dinaric Alps. In contrast, the differences in the spatial precipitation patterns between the full saturation experiment and the control simulation are small (Fig. 5(d)). This is also true for the southern Spain soil condition experiments (not shown). Furthermore, only few small areas of the differences of the two most extreme sensitivity experiments are significant at the 5 % level using a non-parametric Mann–Whitney–U test, indicating also a high variability in the exact location of the precipitation changes within the five cases.

## 4.2   Atlantic SSTs

The sensitivity experiment with increased and decreased SSTs in the Atlantic Ocean reveals only moderate changes in all variables (Fig. 4(f) to (j)), and none of the variables show significant changes compared to the control simulation. For mean daily precipitation in domain 3, there is almost no change detectable with changing Atlantic SSTs. Daily mean moisture flux over land, precipitable water and CAPE show a very small change with decreasing and also increasing Atlantic SSTs compared to the control simulation (Fig. 4(g), (i), (j)). The daily mean moisture flux over the Mediterranean Sea shows an inverse behaviour compared to the rest of the variables, i.e., an increase (decrease) in Atlantic SSTs results in a decrease (increase) of 9 % (7 %) compared to the control experiment (Fig. 4(h)). This is because the impact of the Atlantic SSTs is only indirectly captured. The surface moisture flux over the Atlantic Ocean increases (decreases) with increasing (decreasing) SSTs and thus the atmospheric moisture content becomes more (less) saturated when the air reaches the Mediterranean Sea. Hence, the Mediterranean Sea behaves in the opposite direction as the Atlantic, i.e., a reduced moisture flux over the Mediterranean

Sea is observed as long as the Atlantic Ocean supplies the atmosphere with moisture, and vice versa. Note that the changes in moisture flux over the Mediterranean Sea are still relatively small and indeed insignificant.

This lack of sensitivity to Atlantic SSTs means that precipitation of high-impact summer Vb events hardly changes with changing SSTs. Therefore, also the precipitable water in domain 3 only increases slightly. As moisture content in the atmosphere increases marginally, the latent energy remains almost unchanged and thus, CAPE does not vary between these experiments.

The small observable sensitivity in the mean (Fig. 4) are also evident in the precipitation patterns of the Atlantic SST experiment. The two most extreme sensitivity experiments show on average over the five precipitation-intense summer Vb events in both cases a patchy pattern with insignificant anomalies of both signs throughout domain 3 (Fig. 5(e) and Fig. 5(f)). The insignificance can be explained by a large case-to-case variability in the precipitation changes for the five Vb events selected.

### 4.3 Mediterranean SSTs

An increase (decrease) in the SSTs of the Mediterranean Sea leads on average over the five analysed Vb events to an increase (decrease) in daily mean precipitation, daily mean upward moisture flux over the Mediterranean Sea, in precipitable water and in mean CAPE (Fig. 4(k)–(o)). Particularly, an increase of 5 K in the Mediterranean SSTs leads to a significant increase in precipitation of 24 % on average, while a reduction in Mediterranean SSTs induces a reduction in precipitation of only 9 % compared to the control simulation (Fig. 4(k)) indicating a non-linear relationship further discussed below. The daily mean upward moisture flux over land shows no change over the different Mediterranean Sea sensitivity experiments (Fig. 4(l)). As expected, changes in the Mediterranean SSTs have the strongest impact on the daily mean moisture flux over the Mediterranean Sea compared to the other variables shown in Fig. 4. This is because an increase (a reduction) in SSTs of 5 K results in a change of 124 % ($-65$ %) in the mean moisture flux over the Mediterranean Sea compared to the control simulation (Fig. 4(m)). Besides the daily mean upward moisture flux over land, also precipitable water shows small deviations due to changes in the Mediterranean SSTs compared to the control simulation. Hence, precipitable water increases (decreases) insignificantly by 8 % (4 %) with an increase (a decrease) of 5 K in the Mediterranean SSTs (Fig. 4(n)).

As indicated above, the Mediterranean SST sensitivity experiments exhibit a non-linear increase in precipitation amounts in domain 3 with increasing SSTs (Fig. 4(k)). This can be due to two different mechanisms. One is the increased moisture flux induced by increased SSTs. This increased moisture flux leads to a mostly linear increase in the average atmospheric moisture, as demonstrated by the amount of precipitable water in Fig. 4(n). Nevertheless, the non-linear behaviour observed in the average precipitation is driven by an increase in atmospheric instability, i.e., CAPE. Hence, an increase in atmospheric water vapour goes along with an increase in latent heat and leads to additional convection, which is capable of removing an even larger portion of water than expected from the single increase in atmospheric moisture.

As expected from the distinct changes described above, Mediterranean SST variability leads to significant anomalies in the average precipitation pattern for the +5 K experiment (Fig. 5(h)). The experiments with +1 to +3 K show almost no significance, whereas the +4 K experiments show similar significance patterns as the +5 K experiment, but with a smaller amplitude (not

shown). The cooling experiments, including the -5 K experiment, do not generate significant changes on the 5 % significance level (Fig. 5(g)) compared to the control simulation (Fig. 5(b)). For the sensitivity experiments with the Mediterranean SSTs an increase in SSTs leads to a strong increase in precipitation over coastal areas, together with a reduction in precipitation over the Alpine areas. This is explained by the loss of moisture over the coastal areas in the sensitivity experiments induced by the destabilisation of the atmosphere pointed out above. Note that the changes over the coastal areas are not significant, since the exact location and amount of precipitation varies across the five high-impact summer Vb events. This increased precipitation is responsible for the removal of great amounts of atmospheric moisture so that the precipitation over Central Europe, and especially the Alps, is reduced as a side effect. The significant pattern in precipitation reduction nicely resembles the water transport towards the Alps that is significantly reduced for the +5 K Mediterranean SST experiment. In case of a cooling, there is a reduced precipitation over coastal areas because of an increased stability of the atmosphere. Since the precipitation is reduced in coastal areas, the air is more likely saturated when it hits the Alps during the Vb event. Hence, more precipitation can fall in the Alpine region during the event with decreased SSTs in the Mediterranean Sea. However, such changes for a decrease in Mediterranean SSTs are not significant on the 5 % significance level.

## 4.4 Discussion

The three families of sensitivity experiments suggest that the analysed Vb events are mostly sensitive to changes in the Mediterranean Sea and seem to be rather insensitive to changes in the Atlantic SSTs and the soil moisture content. This is because an increase of 5 K in the Mediterranean SSTs leads to a rise in precipitation of up to 24 % over Central Europe. This high number can otherwise only be exceeded by an initialized and complete desaturation of the soil moisture in whole domain 1 and all four layers of the Noah soil model implemented within WRF. However, the latter experiment is an unrealistic extreme and more realistic situations are not likely to provoke an appreciable impact on the severity of precipitation-intense summer Vb events, as the southern Spain soil condition experiment confirms. Furthermore, the insensitivity of the analysed Vb events to Atlantic SST changes, might also be due to the fact that they are all observed during summer. This is consistent with the argument that the North Atlantic might influence the atmospheric moisture more strongly in winter (Sodemann and Zubler, 2010). Still this does not mean that the Atlantic has no influence on the Vb cyclones throughout a season as seasonal SST change might change the atmospheric circulation stimulating the generation of Vb cyclone. Nevertheless, such responses are not possible to asses with the experimental design selected and are thus beyond the scope of the study.

Our results are in line with the case studies of Sodemann et al. (2009) and Gangoiti et al. (2011) as they identified the Mediterranean basin as a key area for the massive precipitation over Europe during the Vb event in August 2002. Sodemann et al. (2009) additionally suggested that the moisture sources during this event include the Atlantic Ocean, evapotranspiration from land areas, and long-range advection from subtropical areas outside the model domain. However, the latter results can only partially be confirmed in our study, since we found only marginal contributions of soil moisture and Atlantic SST changes to precipitation amounts. Still, our study cannot be directly compared to the results found by Sodemann et al. (2009), since we summarise the main moisture source from various high-impact summer Vb events instead of one isolated case study. Additionally, our analyses are also in line with results obtained from GCM simulations showing an amplification of extreme summer

precipitation by rising Mediterranean SSTs from the period 1970–1999 to the period 2000–2012 (Volosciuk et al., 2016). Furthermore, our study and Volosciuk et al. (2016) seem to agree on the reduction in precipitation over eastern Switzerland and western Austria. Besides this, evaporation from land is frequently identified as an important moisture source during Vb events, as found by Ulbrich et al. (2003a) and Stohl and James (2004) for the Vb event in 2002, and by Grams et al. (2014) and Kele-

men et al. (2016) for the Vb event in 2013. The 2013 Vb event is not included in our study, because it follows a rather untypical Vb trajectory. This might be one reason for the different result in this study and the ones carried out by Grams et al. (2014) and Kelemen et al. (2016). Furthermore, only the soil moisture volume at the beginning of the event is artificially removed, thus allowing moisture recycling during the event. This might be an additional reason for the divergence in the results on moisture evaporation from land. Nevertheless, it is important to emphasise that the main difference between this study and the studies

mentioned above is that we analyse the main driving moisture source of different Vb events instead of a single case study. Thus, it cannot be expected that the average behaviour of several Vb events fully agrees with single case studies. Even though the agreement between these events is relatively large, there is still case-to-case variability. Additionally, it is noteworthy that the fact that the Mediterranean Sea seems to be the main contributor to heavy precipitation events independent of case studies is in line with Gimeno et al. (2010).

Furthermore, the increase in precipitation in coastal areas as they were found for the Mediterranean SST experiment is confirmed by the study of Meredith et al. (2015). In their study they attributed a strong increase at the eastern coast of the Black Sea to increases in the SSTs of the Black Sea. Meredith et al. (2015) also argued that the strong increase in precipitation is connected to an enhancement of the instability in the lower troposphere that allows to trigger deep convection.

## 5  Analysis and discussion of changes in cyclone (and) characteristics

Since the Mediterranean Sea seems to be the most important factor for the analysed high-impact summer Vb events, this section focuses on the sensitivity of the dynamics of the cyclones in the experiments with the Mediterranean SSTs.

The ten tracks (stippled lines in Fig. 3) obtained by the sensitivity experiments with the Mediterranean SSTs for each of the five studied Vb events line up with the tracks obtained in the control simulation (light green line in Fig. 3). Especially the first time steps of each of the events show a good agreement between the ten sensitivity experiments and the control simulation.

Only during the mature and decaying phase of the cyclones, the tracks within the sensitivity experiment start to diverge (Fig. 3). This indicates that deviations in the track cannot be made responsible for changes in the precipitation within the sensitivity experiments. A strong latitudinal displacement of the tracks might have influenced and changed the moisture advection to the impact area over Central Europe and hence, precipitation amounts. Since only very small deviations within the tracks are found this effect can be excluded.

Another important variable for the dynamics of a cyclone is the mean gradient within an area of $1000 \times 1000$ km$^2$ at 850-hPa which is a measure of the wind intensity around a cyclone assuming the geostrophic approximation. The analysis shows that the cyclone with the steepest gradient during its life time is almost insensitive to changes in the Mediterranean SSTs (Fig. 6(a)). In contrast, the cyclone that has the weakest gradient of the five studied Vb events shows a much stronger sensitivity to changes

in the Mediterranean SSTs (Fig. 6(b)). Thus, a warming of the Mediterranean SSTs has the potential to intensify Vb cyclones, while a slight reduction in intensity can be obtained by cooling the Mediterranean SSTs. The three other analysed cyclones (not shown) obtain maximum gradients located in between the ones depicted in Fig. 6. Therefore, it seems that the five analysed summer Vb cyclones show an increasing sensitivity towards changes in the Mediterranean SSTs with decreasing maximum

gradient. This is especially true during the first 30 to 50 hours of the life time of a cyclone, i.e., during the intensification phase. These results may indicate that a maximal threshold of the cyclone is reached in the most intense one, so only weaker cyclones are able to intensify with warmer Mediterranean SSTs. This threshold can be interpreted as energy threshold as the gradient in 850-hPa geopotential height around a cyclone is related to the wind speed (via the geostrophic approximation) and thus the kinetic energy. Therefore, our results indicate that warmer Mediterranean SSTs lead in a non-linear way to stronger kinetic

energy, whereas the growth of the strongest cyclones might be capped by a possible upper energy limit. This result is in line with the work of Pepler et al. (2016) on southern hemispheric cyclones. They investigated the influence of eastern Australian coastal SSTs on extra-tropical cyclone intensification and results suggest that SSTs play only a minor role in the intensification of the most intense cyclones, as they are more strongly influenced by the prevailing atmospheric conditions. Also the work of Blender et al. (2016), who analysed extreme values in vorticity and geopotential height (GPH) fields during the winter, support

that extremes in the GPH might be limited by an upper bound.

## 6   Summary

In this study, we identify the main moisture source for a composite of five different high-impact summer Vb event. For this three different families of idealised sensitivity experiments are carried out over five precipitation-intense summer Vb events that occurred in the period between 1979 to 2013. The three sensitivity experiments include artificial removal and supply of

soil moisture as well as changes in the SSTs of the Atlantic Ocean and the Mediterranean Sea. The experiments are conducted with the regional model WRF, driven with the ERA-Interim reanalysis dataset.

The validation of WRF with two observational data sets, E-OBS and EURO4m-APGD, reveals that WRF is generally able to reproduce precipitation amounts in Vb events over the Alpine region. There is however a slightly better agreement with EURO4m-APGD, which suggests that the convective processes largely responsible for summer precipitation in the Alps are

reasonably reproduced by the model. Hence, the latter database seems to be more suitable than E-OBS for recording the precipitation in this area of complex topography.

Additionally, the track characteristics of the high-impact summer Vb events in the control simulations exhibit a good agreement with the ones obtained in the ERA-Interim dataset. This allows gaining faith in the model's ability to simulate the relevant physical processes in a reasonable way.

Various sensitivity experiments are carried out, which allow drawing the following conclusions: A complete removal of the soil moisture content over great parts of Europe and in all four layers of the soil model in the initial conditions leads to a notable reduction in daily mean upward moisture flux over land, which leads to an increase in sensible heat flux and a reduction in latent heat flux. The increase in sensible heat conversely drives a reduction in relative humidity. The reduction in daily mean

upward moisture flux and relative humidity lead to a reduction of approximately 32 % in precipitation over Central Europe. For the southern Spain soil condition sensitivity experiment the processes just described are valid but in a smaller extent such that the reduction in precipitation only reaches 7 %. Conversely, for an increase in soil moisture content the same processes hold but in the inverse and also in a reduced way, and hence it leads to a small increase of around 7 % in precipitation.

Nevertheless, these soil moisture experiments but especially the complete drainage experiment are very unrealistic and extreme. Still, it seems unlikely that a considerable impact on the severity of precipitation-intense summer Vb events, i.e. on precipitation amounts, can be obtained in more realistic scenarios.

The changes in precipitation patterns for the soil moisture experiment show generally a decrease (increase) over domain 3 for a full drainage (saturation) of the soil moisture content. Nevertheless, the case-to-case variability for the location of the

precipitation changes is high and inconsistent, and thus no significant changes are found ($p < 0.95$).

Similarly, the sensitivity experiments varying the Atlantic SSTs show almost no change in precipitation over domain 3, indicating that on average the analysed Vb events are hardly sensitive to changes in the Atlantic SSTs. The same holds true for the precipitation pattern changes for the Atlantic Ocean. In these experiments the sign and location of changes varies between single Vb events, and hence no significant change can be found, neither for increasing nor for decreasing Atlantic SSTs.

A 5-K increase in the Mediterranean SSTs leads to a similar absolute change in precipitation than a complete removal of the soil moisture content. Hence, an increase in Mediterranean SSTs of 5 K leads to an increase in precipitation of approximately 24 %. The larger precipitation rates for warmer Mediterranean SSTs are induced by a strong increase in daily mean upward moisture flux over the Mediterranean Sea, together with a decrease in the atmospheric stability induced by the release of more latent heat. While the increase in mean upward moisture flux feeds a linear increase in precipitable water, i.e., the water content

in the atmosphere, a non-linear increase in CAPE, i.e., the atmospheric instability, leads to convection that is able to remove more moisture from the atmosphere than expected by a single increase in water vapour. Hence, a non-linear behaviour is found in the precipitation sensitivities, attributable to an increase in atmospheric instability with increasing Mediterranean SSTs due to a strong significant increase in moisture flux over the Mediterranean Sea. Conversely, a decrease in Mediterranean SSTs leads to inverted processes as those described before, and thus produces a slight reduction in precipitation over Central Europe.

The increase in Mediterranean SSTs by 5 K generates changes in the Balkan coastal areas together with significant decreases in precipitation amounts over the eastern ridge of the Alps. This indicates that the air contains enough moisture to precipitate out while it is lifted over the Dinaric Alps. Note that the exact location and amount of precipitation does change within the different Vb events, and consequently no significant change can be obtained here. This topographic-induced precipitation leaves the air drier than in the control experiment when it reaches the Alpine area, and explains the significant reduction in precipitation over

the whole expected air advection path of a Vb event. The same mechanism, but reversed, happens in a cooled Mediterranean SSTs scenario. Still, unlike in the former case, the changes induced by a cooling of the Mediterranean SSTs do not reach a significant level ($p < 0.95$).

The above-mentioned changes in precipitation amounts and patterns indicate, from all the sensitivities analysed, that these five analysed precipitation-intense summer Vb events are mostly sensitive to changes in the Mediterranean SSTs.

The Mediterranean SST experiments allow further interesting findings. While there is a good agreement in the trajectories of Vb events across sensitivity experiments, the intensity measured by gradient within an area of $1000 \times 1000$ km$^2$ around the cyclone centre is generally different in the various sensitivity experiments carried out. In particular, we found that a warming of the Mediterranean SSTs can lead to an increase in the gradient, and thus to a more intense cyclone during its intensification period within the first 30 to 50 hours. Similarly, a decrease in the cyclones intensity is found for a decrease in Mediterranean SSTs. Interestingly, the change in intensity of the cyclone is inversely proportional to the maximal intensity that is obtained during a cyclone's life time in the control experiment. This is, the most intense cyclone shows little to no change in intensity, neither for decreasing nor for increasing Mediterranean SSTs. This may indicate that strong cyclones are limited in growth of kinetic energy since they might be capped by an upper bound. On the other hand there seems to be the possibility for weaker cyclones to grow in kinetic energy with increasing Mediterranean SSTs in a non-linear way. A possible reason for the limited sensitivity of strong cyclones to changes in the intensity might be, that these cyclones are more strongly steered by the large-scale atmospheric conditions, as described by Pepler et al. (2016).

As a final remark these results shall not be understood as climate change projections. An important drawback in this type of sensitivity studies is that to some extent the physical consistency cannot be granted. In our setup, the most non-physical problem is the heating of the ocean surface alone. This has the effect that a strong and artificial temperature gradient is introduced near the coastal areas, which does not correspond to a natural behaviour. Although in these experiments the model seems to bring this disturbance back to a physically plausible situation after a few hours, this introduces artefacts in the simulation, which are difficult to isolate. Therefore, obtaining more physically consistent and thus reliable results would require running transient simulations driven by comprehensive Earth System Models under realistic climate change scenarios.

## 7 Data availability

Data is available upon request from the corresponding author Martina Messmer (messmer@climate.unibe.ch).

*Author contributions.* Martina Messmer, Juan José Gómez-Navarro and Christoph C. Raible contributed to the design of the experiments, Martina Messmer ran the simulations and wrote the first draft. All authors contributed in the internal review of the text previous to the submission.

*Competing interests.* The authors declare that they have no conflict of interest.

*Acknowledgements.* The authors are grateful for the funding provided by the Dr. Alfred Bretscher-Fonds für Klima- und Luftverschmutzungs-forschung. Thanks are also due to the support provided by the Oeschger Centre for Climate Change Research and the Mobiliar lab for climate risks and natural hazards (Mobilab). Juan José Gómez-Navarro acknowledges the funding provided through the contract for the return of experienced researchers, resolution R-735/2015 of the University of Murcia and the CARM for the funding provided through the Seneca

Foundation (project 20022/SF/16). The ERA-Interim reanalysis data were provided by the ECMWF. Furthermore, we acknowledge the E-OBS data set from the EU-FP6 project ENSEMBLES (http://ensembles-eu.metoffice.com) and the data providers in the ECA&D project (http://www.ecad.eu). Thanks are due to European Reanalysis and Observations for Monitoring for providing us with the APGD dataset. The simulations are all run at the Swiss National Supercomputing Centre CSCS. Thanks are due to the two anonymous referees and the Editor Rui A. P. Perdigão for their constructive comments that helped to improve the manuscript.

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

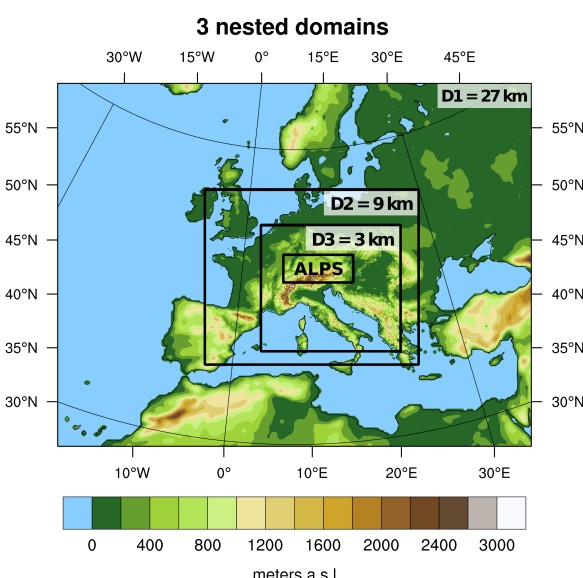

**Figure 1.** The three nested domains (D1 to D3) with their actual resolution are depicted as black boxes. The box labelled "Alps" denotes the area used for measuring the precipitation intensity of the Vb events. The shading shows the topographical elevation implemented in the simulations in meters above sea level.

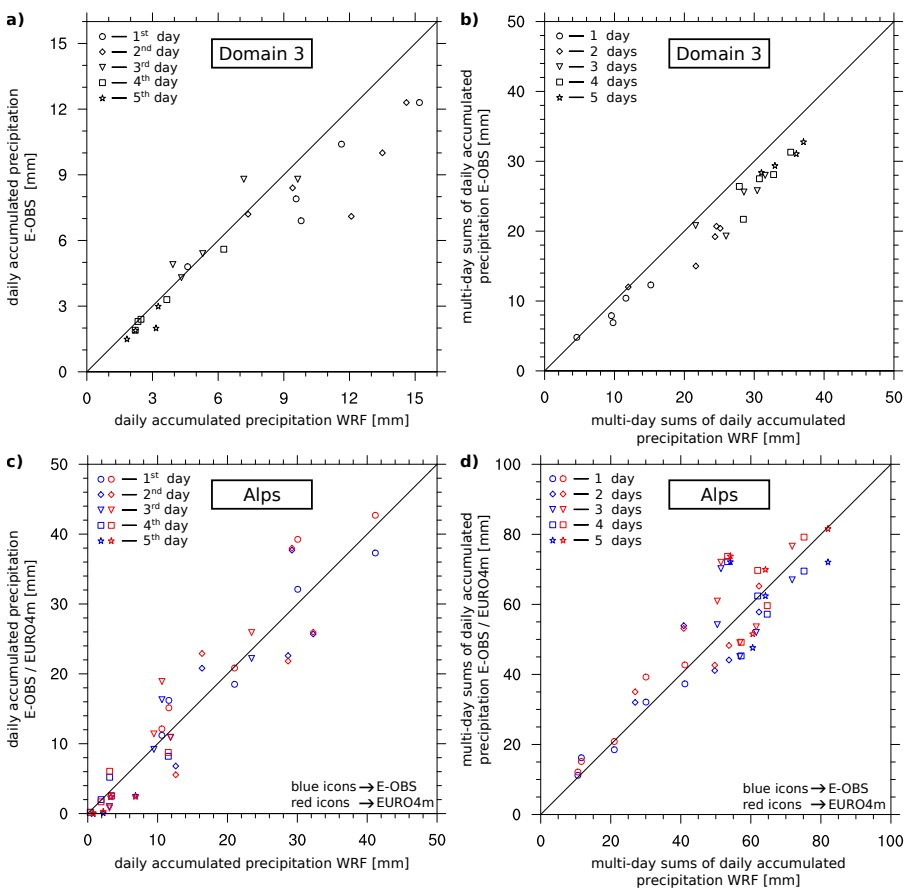

**Figure 2.** The left column shows the daily accumulated precipitation [mm] obtained by observations plotted against the one obtained by WRF for (a) domain 3 and (c) the Alps ("Alps" box in Fig. 1) for each of the five days of the five different Vb events. The right column depicts the multi-day sums o daily accumulated precipitation [mm] for 1 to 5 days for the observations against the one obtained by the WRF simulations for (b) domain 3 and (d) the Alps for each of the five analysed Vb events. The upper row uses E-OBS as observational data set, while the bottom row depicts E-OBS (blue icons) and the EURO4m-APGD (red icons) as observational data sets.

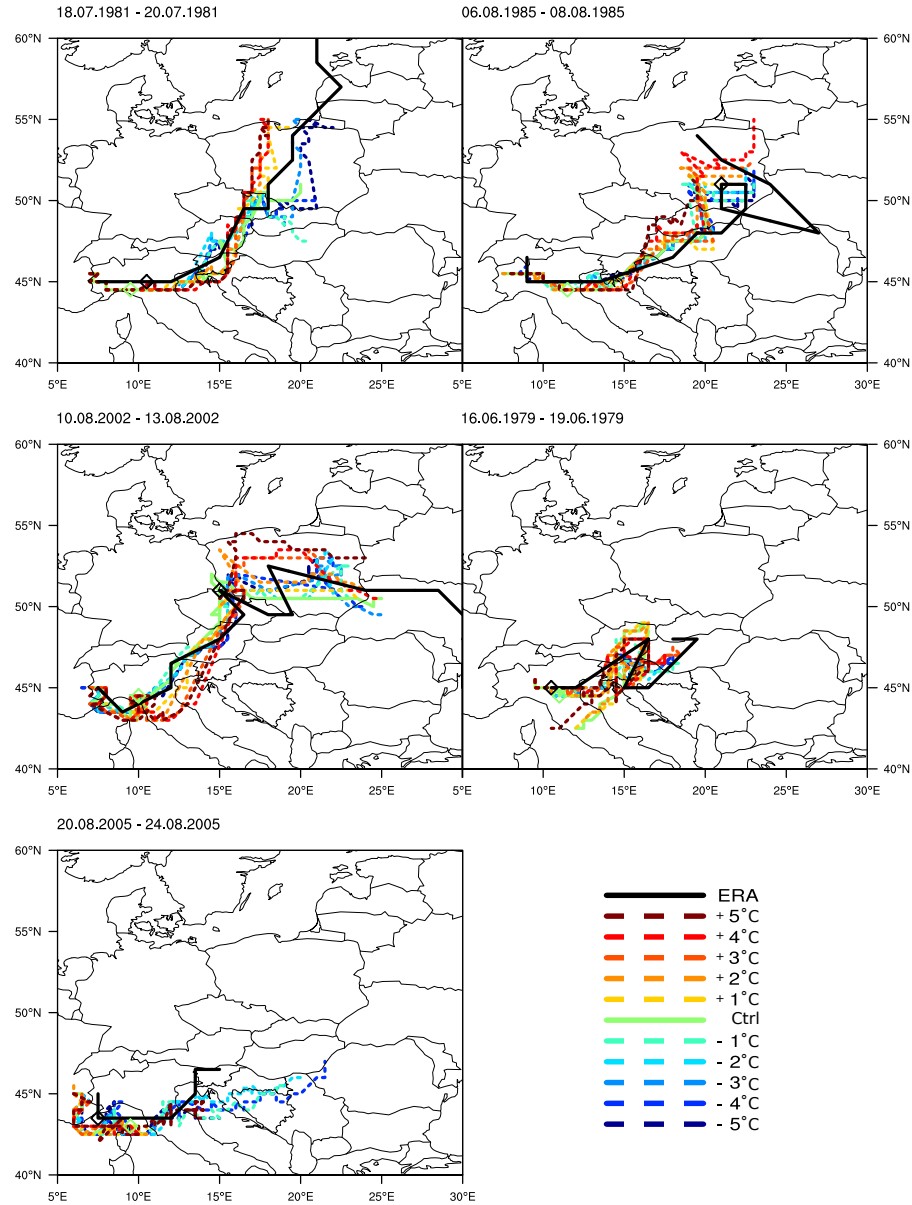

**Figure 3.** Tracks for the five different analysed Vb events. The black line depicts the tracks that are obtained using the ERA-Interim dataset. The light green line shows the tracks detected in the control simulation. The stippled lines show the tracks of the different Mediterranean SST experiments. The green and the black diamond represent the point of the cyclone, at which it reaches the strongest gradient during its lifetime for the control simulation and ERA-Interim, respectively.

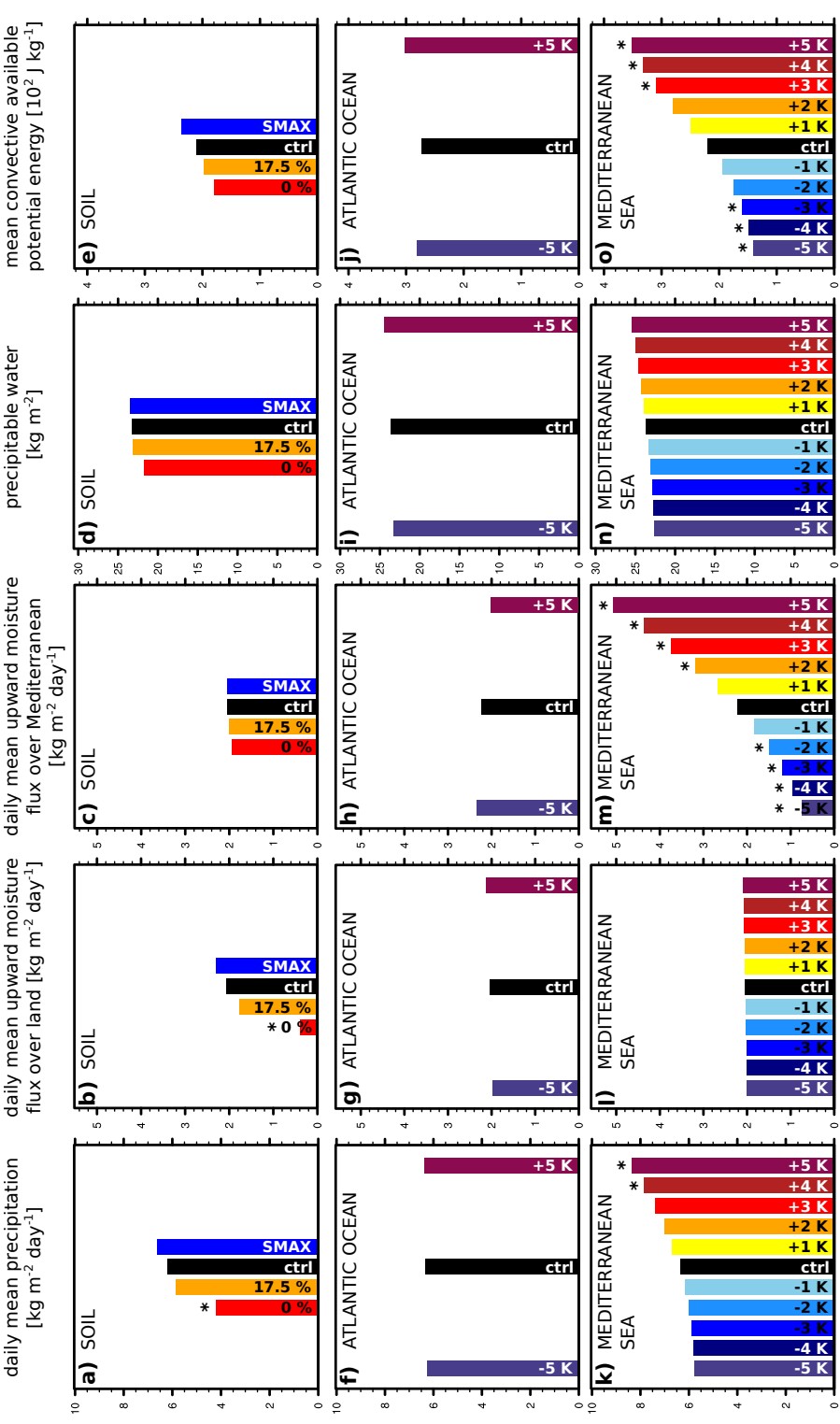

**Figure 4.** The panels in the first row show the mean over 5 Vb events for the soil experiments with a bar for drainage (0 %, red), southern Spain soil water conditions(17.5 %, orange), the control simulation (ctrl, black) and full saturation (SMAX, blue). The second row shows the mean over 5 Vb events for the Atlantic SST experiments with bars depicting a decrease in SSTs of 5 K on the left to an increase in SSTs of 5 K on the right (red). The third row shows the mean over 5 Vb events for the Mediterranean SST experiments with bars depicting a decrease in SSTs of 5 K on the left to an increase in SSTs of 5 K on the right with increments of 1 K. The five columns show the daily mean precipitation, upward moisture flux over land and over ocean, and mean convective available potential energy (CAPE) for D3 from the left to the right. Stars above the bars denote significant changes compared to the control simulation using a non-parametric Mann–Whitney–U test and the 5 % significance level. The units for the y-axis are given in the header of each column, whereas the x-axis denotes the performed sensitivity studies.

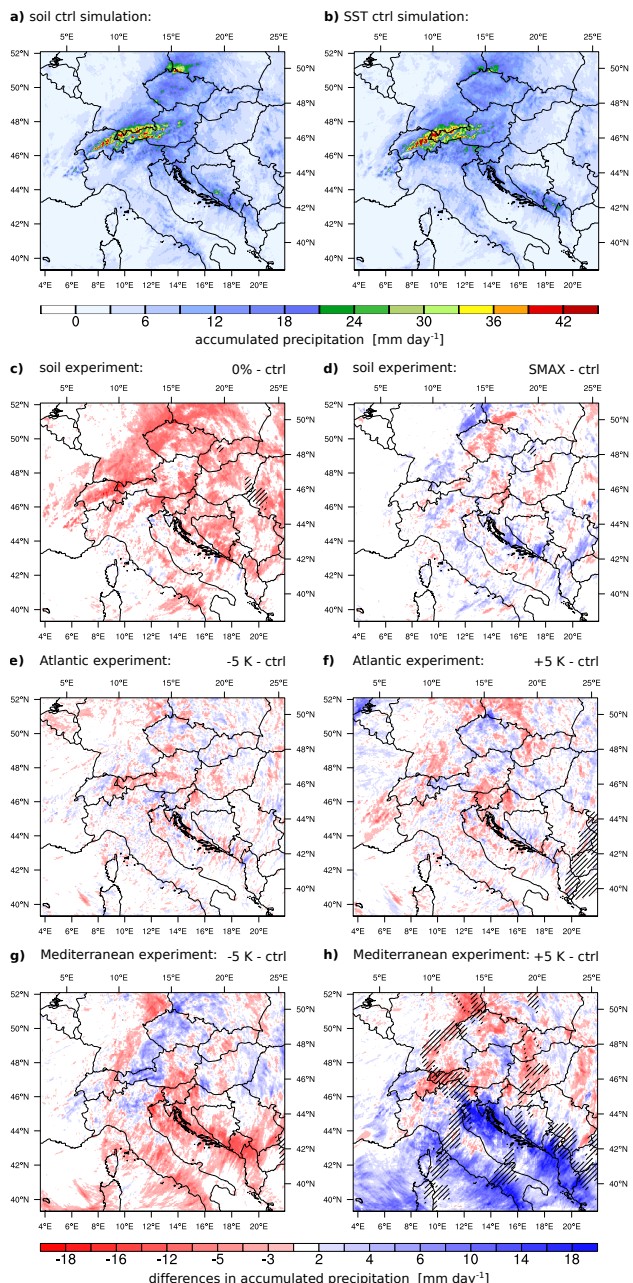

**Figure 5.** Panel (a) shows the accumulated precipitation [mm day$^{-1}$] for the control simulation of the soil moisture experiments averaged over the five analysed Vb events. Panel (b) shows the same as panel (a) but for the SST experiments. The second to the fourth row show the differences between the mean daily precipitation obtained by the different sensitivity experiment and the control simulation [mm day$^{-1}$]. (c) shows the complete drainage soil experiment, (d) the full saturation soil experiment, (e) and (f) the -5 and +5 K Atlantic SST experiment, respectively, (g) and (h) the -5 and +5 K Mediterranean SST experiment. The hatched area denotes significant changes at the 5 % significance level using a non-parametric Mann–Whitney–U test.

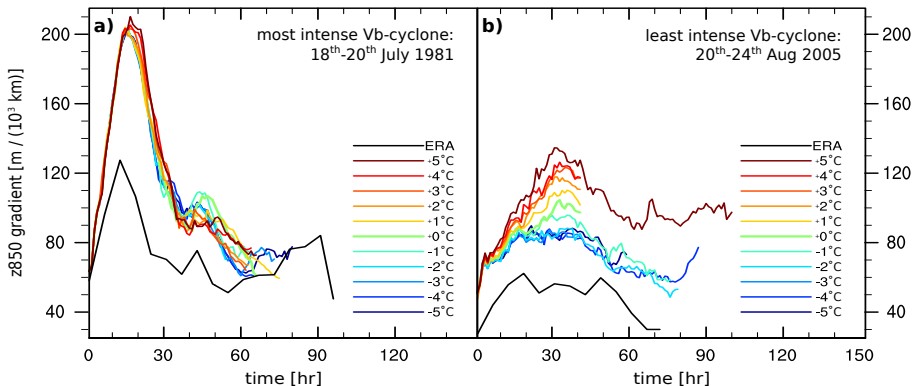

**Figure 6.** The gradient within an area of $1000\times1000$ km$^2$ for the geopotential height at 850 hPa is shown for two different Vb events. The coloured lines indicate changes in the gradient over time of the Mediterranean SST experiments. The black line shows the evolution in the gradient in the ERA-Interim data for the same event. On the left panel the most intense analysed Vb event (18th–20th July 1981) is shown. On the right, the least intense of the analysed Vb events (20th–24th August 2005) is shown. The data based on WRF shows an hourly resolution and a spatial resolution of $0.5°\times0.5°$. The ERA-Interim data is based on 6-hourly temporal resolution, which has been linearly interpolated on 1-hourly temporal resolution, while the spatial resolution is $1.5°\times1.5°$.

**Table 1.** Important parameterisations used to run the WRF sensitivity experiments.

| Parameterisation | Parameter name | Chosen parameterisation | Applied to |
|---|---|---|---|
| Microphysics | mp_physics | WRF single moment 6-class scheme | Domain 1–3 |
| Longwave radiation | ra_lw_physiscs | RRTM scheme | Domain 1–3 |
| Shortwave radiation | ra_sw_physics | Dudhia scheme | Domain 1–3 |
| Surface layer | sf_sfclay_pysics | MM5 similarity | Domain 1–3 |
| Land/water surface | sf_surface_physics | Noah Land Surface Model | Domain 1–3 |
| Planetary boundary layer | bl_pbl_physics | Yonsei University scheme | Domain 1–3 |
| Cumulus | cu_physics | Kain–Fritsch scheme | Domain 1 |
| | | Grell–Freitas scheme | Domain 2 |
| | | No parameterisation | Domain 3 |