# Peer review of "Sensitivity Experiments on the Response of Vb Cyclones to Sea Surface Temperature and Soil Moisture Changes"

_Earth System Dynamics, 2016_

## Referee Comment (RC1) · Anonymous Referee #1 · 22 Dec 2016

General comments:

Messmer et al. examine the influence of sea surface temperature (SST) and soil moisture on precipitation and cyclone characteristics for 5 selected historical Vb cyclones. They conclude that the main factor influencing cyclone strength and the amount of precipitation is the Mediterranean SST. The subject is is of scientific interest and the chosen methodology is suitable for the analysis.

My main concern for this paper is that a very similar study already exists, that the authors don't seem to be aware of: Volosciuk, C. et al. Rising Mediterranean Sea Surface Temperatures Amplify Extreme Summer Precipitation in Central Europe. Sci. Rep. 6, 32450; doi: 10.1038/srep32450 (2016). It is important that the authors cite this

study and highlight how their study differs from the one of Volosciuk et al. They need to compare the results and discuss similarities and differences.

Another point they should address is the following: In their last paper "Climatology of Vb cyclones, physical mechanisms and their impact on extreme precipitation over Central Europe" they conclude that heavy precipitation related to Vb events is mainly related to large-scale dynamics rather than to thermodynamic processes, yet they decide to analyse the effect of changes in SSTs. This needs some motivation.

Specific comments:

Introduction: The only uplifting process mentioned is that at the northern side of the Alps. The Central European part of the precipitation that caused the Central European floods of 2002 and 2013 also involved uplifting at low mountain ranges such as the "Erzgebirge". Reading the paper one gets the impression that only the Alps are important.

p.4 line 21: "2-way basis" are you speaking about 2-way nesting in which the higher resolution results feed back on the lower resolution? Please explain further.

p.5 line 4: Does each domain has a time lag of 6 hours for initialization compared to the next bigger domain, thus an accumulated initialization of 18 hours to ERA-Interim?

p.6 line 6: For which scenario(s)? How do SSTs relate to surface air temperature?

p 6 line 18: Please explain in more detail. Do you interpolate both values to the same grid? Do you show the mean rate over the entire area? E-OBS doesn't have values over the ocean, however, domain D3 for which you compare precipitation includes ocean areas. How do you handle this? The cyclones don't remain in the area of interest for 5 days. How and why did you select a 5-day period?

p.6 line 19: If you compare precipitation rates for extreme events you expect that you get different values if you use different grids. You expect lower values for a lower resolution data set as it represents the average over a larger area. (see for example

Göber et al.: Could a perfect model ever satisfy a naiive forecaster? Meteorol. Appl., 15, 359 – 365, 2008)

p.6 line 27: Later you show that there is no indication of a positive bias for WRF for extreme events. With respect to biases you need to distinguish between mean and extreme precipitation.

p.7 line 29: Averaging CAPE and precipitation over the entire domain D3 obscures the signal that can be attributed to the Vb cyclones. How much do your results differ if you average over a smaller domain close to the location of the cyclone?

p.10 line 29-30: Can you confirm the results of Sodemann et al. if you look only at the 2002 case?

p.11 line 30: The statistical basis is too small for this statement.

p.12 line 6: You selected the 5 strongest events, so "average" does not seem to be the appropriate term.

p.13 line 30: Some information on "maximum energy" concept for cyclones is needed here. What does it mean? How is the energy of a cyclone determined? Do you want to imply that no cyclones stronger than the selected one can appear in this region? This is a strong statement that you should check for the ERA-Interim resolution cyclones in that region.

p.13 line 31: "strong cyclones are strongly steered by the atmospheric conditions" (This sentence is also included in the abstract). I don't understand what you are trying to say here and how it is a consequence of the previous statement. You need to explain this in more detail.

Technical corrections:

p.2 line 4: Vb cyclones are not characterized as cyclones they are cyclones.

p.2 line 5-6: As they reach .... they turn ....

P.4 line 30: "The increase of the Atlantic Ocean SSTs is guided by the expected changes in the Mediterranean SSTs described in Sect. 2.4.3." I don't understand this sentence. Which changes did you expect for the Mediterranean SSTs in response to the Atlantic SSTs?

p.5 line 4: "The SSTs that are deviant compared to the control simulation are then prescribed after the vertical interpolation step of 5 meteorological data onto the domain grid." I don't understand what was done. I am surprised vertical interpolation is needed as all regionalizations use the same 50 vertical levels. I thought were adding a constant to the control SSTs. Why do you need to determine anomalies from the control? Please rephrase.

p.11 line 13: Analysis and discussion of changes in cyclone (and) characteristics

p.13 line 33: consistency not inconsistency

———————————————

---

## Referee Comment (RC2) · Anonymous Referee #2 · 13 Jan 2017

The study analyses the moisture sources of 5 selected Vb cyclones, which caused high precipitation over the Alpine region. To investigate the moisture sources, sensitivity studies are used, with altered SST of the Mediterranean Sea, the North Atlantic Ocean and soil moisture. They found that the analysed events were most sensitive to the Mediterranean SST changes, and although the SST increase resulted in a strong increase of precipitation over the region influenced by the Vb cyclones, it also resulted in a decrease of precipitation over the north-eastern flanks of the Alps. The study is well structured, and deals with a scientifically interesting subject.

General comments

The study aims to give a general view on the moisture sources of Vb events, but only

focuses on 5 selected events from summer, which were all connected to heavy precipitation. The extreme cases are indeed interesting, but the study should underline it more that these are not typical Vb evenents, since even their previous study (Messmer et al. 2015) concluded that only 23 % of all Vb events are associated with extreme precipitation. Also their conclusions are valid mainly for summer, due to the event selection. This should be mentioned, since previous studies have shown that moisture sources in the Alpine region is influenced by seasonality, and for example the North Atlantic region is a more pronounced source during winter (Sodemann and Zubler, 2010). Thus the low sensitivity to the changes of North Atlantic SST might not be valid for the whole year.

I found the 6 hour spin-up time rather short. I would expect that the water vapour fluxes do not have enough time to adjust to the altered boundary conditions. Also Winschall et al. (2014) found that for heavy precipitation events over a slightly different domain, the time of maximum moisture uptake varies between a few hours to more than a week before the precipitation event. So with 6 hours spin up time the moisture uptake is probably already occurred, and included in the initial and boundary conditions (SST and soil moisture). So the changed boundary values and thus moisture fluxes have less effect on the cyclonic precipitation. The authors already analysed the sensitivity for the spin-up time in case of the SST experiments, but I would like to ask for more details about those analysis, and also the a revision of the above mentioned moisture uptake "problem". Also I do not see if they have investigated the spin up effect during the soil moisture experiments. I would like to see some results regarding this, because the 6 hour spin-up time also seems to be rather short for the soil experiments.

Specific comments

Abstract: The soil moisture experiments are mentioned, but no results are included, also the North Atlantic experiments are not mentioned.

Page 2 Line 8: Besides the Elbe, it would be nice to mention, which other large rivers

are especially affected by the precipitation of the Vb cyclones.

Page 2 Line 30-32: Please reformulate this sentence so, that is not so strong, it should show that the variability of moisture sources are still high besides the seasonality.

Page 3 Line 8: Please write "extreme Vb events" instead of just "Vb events".

Part 2.1: Can you include here some information about the SST field? If not the ERA Interim SST fields are used for the sensitivity experiments, then please write something about the details of the SST boundary fields at the model setup part.

Part 2.2: Please mention, which variables do you use from E-OBS. Also I found the terminology for precipitation rate a bit misleading. As I understood precipitation rate here means daily accumulated precipitation, and accumulated precipitation means multi-day sums of daily precipitation.

Part 2.3: It might be useful to shortly introduce the synoptic situations regarding the selected events, e.g. what was different and what was the same for the 5 cyclones.

Part 2.4: Can you include more information about, how the atmosphere interacts in the model with the SST boundary conditions and with the soil (e.g. frequency, fluxes).

Page 4 Line 21: Please explain in more detail, what does 2-way basis mean. Is it 2 way nesting?

Page 5 Line 9: Please state clearer that the spectral nudging is done for these extra spin-up sensitivity experiments, otherwise it is a bit confusing after stating that nudging techniques are avoided (Page 4 Line 30).

Page 6 Line 1: Mentioning eleven simulations is misleading, since the control is not considered as a Mediterranean SST sensitivity experiment. So please change to "10 sensitivity and one control" or "10 sensitivity simulations".

Page 6 Line 7: Can you find a projection for SST instead of surface air?

Page 6 Line 23: Higher precipitation rates can also be due to the higher resolution of the simulated data.

Page 7 Line 20: Please denote somehow on the tracks in Fig. 3, which are the intensification and decaying phases of the cyclones.

Page 7 Line 30. The moisture uptake from land and ocean by the cyclones happened probably before the precipitation. I think time steps before the precipitation can also give information about the moisture exchange.

Page 8 Line 17: Why are the ocean-land winds slightly reduced?

Page 9 Line 8: Note that the North Atlantic is shown to be more important during winter precipitation events. So there is maybe a lack of sensitivity because these were summer events.

Page 10 Line 5-16: It would help the understanding if an extra domain, a costal domain, would be introduced, and the results would be visualised in a way similar to Fig. 4k.

Page 10 Line 20: 24% in which direction, and where?

Figure 4c. Moisture flux over ocean, is misleading, since almost all points are from the Mediterranean Sea. It might be clearer if the few North Atlantic Ocean points would be excluded, and the moisture flux would only refer to the Mediterranean Sea.

Figure 6. Please mention the resolution of the different data.

Reference

Sodemann, H., & Zubler, E. (2010). Seasonal and inter-annual variability of the moisture sources for alpine precipitation during 1995-2002. International Journal of Climatology, 30, 947–961. http://doi.org/10.1002/joc.1932

---

## Author Comment (AC1) · 25 Jan 2017

We would like to thank the Anonymous Referee #1 for his positive view and constructive comments on our manuscript on "Sensitivity Experiments on the Response of Vb Cyclones to Ocean Temperature and Soil Moisture Changes". Please note that this is just a short and first reply to the referee's comments. A point-by-point reply will follow with the actual revision process after the decision of the editor.

*My main concern for this paper is that a very similar study already exists, that the authors don't seem to be aware of: Volosciuk, C. et al. Rising Mediterranean Sea Surface Temperatures Amplify Extreme Summer Precipitation in Central Europe. Sci.*

[Figure]

*Rep. 6, 32450; doi: 10.1038/srep32450 (2016). It is important that the authors cite this study and highlight how their study differs from the one of Volosciuk et al. They need to compare the results and discuss similarities and differences.*

Regarding the major concern rose by the reviewer, unfortunately we were not aware of the study by Volosciuk et al. (2016) at the time of writing the first version of the manuscript. This is an interesting and closely related piece of work that we will certainly consider to improve the discussion of our results in the reviewed version. As a brief comment, it is interesting how both studies come to similar results, even if they use quite different methods and motivations. For instance, Volosciuk et al. test the sensitivity of cyclone properties over Central Europe to Mediterranean Sea temperature alone compared to 1970 – 1999 using a GCM, while we use a RCM and consider the sensitivity of Vb cyclones to up to three different variables, i.e. the Mediterranean Sea, the Atlantic Ocean and the soil moisture on 5 extreme Vb events. An additional difference is the fact that we have applied much stronger SST changes of up to 5 K in our study and hence it focuses more on climate change scenarios. Despite these differences, both studies agree on an increase in precipitation taking place in similar regions, making the Adriatic coast the most affected region by an increase in precipitation, while the eastern part of Switzerland and the western part of Austria exhibit a decrease in precipitation in both studies. These agreements are not redundant, but independent evidence that reinforces the findings of both studies. We will point out these similarities and differences throughout the results and discussion sections in the reviewed version.

*Another point they should address is the following: In their last paper "Climatology of Vb cyclones, physical mechanisms and their impact on extreme precipitation over Central Europe" they conclude that heavy precipitation related to Vb events is mainly related to large-scale dynamics rather than to thermodynamic processes, yet they decide to analyse the effect of changes in SSTs. This needs some motivation.*

The fact that we decided to study the response of Vb cyclones to changes in possible moisture sources is that there is still some uncertainty concerning the main moisture source during Vb events. Even though the large-scale dynamics determine if a Vb cyclone delivers heavy precipitation or not, an important source of moisture is still needed to trigger such extreme events. To study all possible sources of moisture, and their possible impact on Vb events, under climate change scenarios, we decided to perform these sensitivity studies. As the motivation seems to be somewhat unclear in the first manuscript, we will clarify this in the new version of the introduction.

Of course, we will also address the specific comments in more details throughout the review process. These comments will certainly help to increase the readability and understandability of this paper. In a point-to-point response we will show how these suggestions and comments will be implemented.

---

## Author Comment (AC2) · 25 Jan 2017

We would like to thank the Anonymous Referee #2 for his careful review of our manuscript on "Sensitivity Experiments on the Response of Vb Cyclones to Ocean Temperature and Soil Moisture Changes". Please note that this is just a short and first reply to the referee's comments. A point-by-point reply will follow with a reviewed manuscript after the decision of the editor.

*The study aims to give a general view on the moisture sources of Vb events, but only focuses on 5 selected events from summer, which were all connected to heavy precipitation. The extreme cases are indeed interesting, but the study should underline it*

[Figure]

*more that these are not typical Vb events, since even their previous study (Messmer et al. 2015) concluded that only 23 % of all Vb events are associated with extreme precipitation. Also their conclusions are valid mainly for summer, due to the event selection. This should be mentioned, since previous studies have shown that moisture sources in the Alpine region is influenced by seasonality, and for example the North Atlantic region is a more pronounced source during winter (Sodemann and Zubler, 2010). Thus the low sensitivity to the changes of North Atlantic SST might not be valid for the whole year.*

The first comment on the characteristics of our chosen Vb events is an important point. Apparently we did not state clearly enough that the study focuses on summer and on extreme events only. This is somewhat a limitation that we will certainly emphasis further in the reviewed manuscript. Still, we note that the emphasis in these cases is not arbitrary. Vb events are frequently associated to high impact events that cause important economical and personal damage in central Europe, being this an important motivation to focus on those events. We will also discuss the point that the reviewer makes on the sensitivity to changes in the North Atlantic SST.

*I found the 6 hour spin-up time rather short. I would expect that the water vapour fluxes do not have enough time to adjust to the altered boundary conditions. Also Winschall et al. (2014) found that for heavy precipitation events over a slightly different domain, the time of maximum moisture uptake varies between a few hours to more than a week before the precipitation event. So with 6 hours spin up time the moisture uptake is probably already occurred, and included in the initial and boundary conditions (SST and soil moisture). So the changed boundary values and thus moisture fluxes have less effect on the cyclonic precipitation. The authors already analysed the sensitivity for the spin-up time in case of the SST experiments, but I would like to ask for more details about those analysis, and also the a revision of the above mentioned moisture uptake "problem". Also I do not see if they have investigated the spin up effect during*

*the soil moisture experiments. I would like to see some results regarding this, because the 6 hour spin-up time also seems to be rather short for the soil experiments.*

We understand the second concern of the referee on the relatively short spin-up of 6 hours. Indeed, this is the reason why we carried out an extensive analysis on this regard, and also included some paragraphs on this issue in our study. In short, our analysis allows us to conclude that this period is long enough, although we perhaps have to describe more profoundly how we came to this conclusion. Therefore, we will further discuss to include an additional figure addressing the spin-up time.

Nevertheless, we did not use longer spin-up times for the soil moisture experiments for a certain reason. This is, we would like to see if reduced soil moisture, as this might happen in a future climate, does influence the precipitation during Vb events. To get a strong enough signal we decided to run two extreme experiments, where we removed and saturated the soil moisture completely in all four model layers. Increasing the spin-up time would lead to a filling of the soil moisture volume and thus the sensitivity of Vb cyclone to soil moisture cannot be analyzed. This is at least true for the first model soil layer, which is the most weather-relevant layer. Maybe the experimental design of the soil moisture sensitivity simulation was not presented in a clear way, so we will clarify this in a revised version.

We will certainly address the specific comments throughout the review process, as these will help to increase the readability and consistency of our study. In a point-to-point response we will show how these suggestions and comments will be implemented.

---

## Author Response (AR1)

**Detailed answer to Anonymous Referee #1**

**General comments:**

*My main concern for this paper is that a very similar study already exists, that the authors don't seem to be aware of: Volosciuk, C. et al. Rising Mediterranean Sea Surface Temperatures Amplify Extreme Summer Precipitation in Central Europe. Sci. Rep. 6, 32450; doi: 10.1038srep32450 (2016). It is impdgrayortant that the authors cite this study and highlight how their study differs from the one of Volosciuk et al. They need to compare the results and discuss similarities and differences.*

As indicated in the previous answer to the referee, we were unfortunately not aware of this study at the time of writing the first draft of the manuscript. We have now included a comparison of the two studies in the discussion part as follows:

Additionally, our analysis are also in line with results obtained from GCM simulations showing an amplification of extreme summer precipitation by rising Mediterranean SSTs from the period 1970–1999 to the period 2000–2012 (Volosciuk et al., 2016). Furthermore, our study and Volosciuk et al. (2016) seem to agree on the reduction in precipitation over eastern Switzerland and western Austria.

*Another point they should address is the following: In their last paper "Climatology of Vb cyclones, physical mechanisms and their impact on extreme precipitation over Central Europe" they conclude that heavy precipitation related to Vb events is mainly related to large-scale dynamics rather than to thermodynamic processes, yet they decide to analyse the effect of changes in SSTs. This needs some motivation.*

The motivation was probably not fully clear. For this reason we have included some more details on this:

Furthermore, the large-scale dynamics seem to determine, if a Vb cyclones delivers high precipitation or not (Messmer et al., 2015). Despite this fact, an important moisture source needs to supply the atmosphere with the required moisture. In fact, these thermodynamical processes, and especially the moisture sources, remain unclear as described in the following.

**Specific comments:**

*Introduction: The only uplifting process mentioned is that at the northern side of the Alps. The Central European part of the precipitation that caused the Central European floods of 2002 and 2013 also involved uplifting at low mountain ranges such as the "Erzgebirge". Reading the paper one gets the impression that only the Alps are important.*

We understand the point of the reviewer and specified this more clearly throughout the section of the Vb event description in the Introduction, hence we stated:

A prominent phenomenon of regional high-impact weather in Central Europe, and especially over the northern ridge of the Alps and the adjacent flatlands and low mountain ranges, is the so-called Vb cyclone. These cyclones transport large amounts of atmospheric moisture to the northern side of the Alps and Central Europe, thus triggering extreme precipitation events (Messmer et al., 2015) and exhibit a great potential for floods in the Elbe catchment (Nied et al., 2014) and the Alpine area including adjacent flatlands and low mountain ranges (e.g., chapter

5 in MeteoSchweiz, 2006).

*p.4 line 21: "2-way basis" are you speaking about 2-way nesting in which the higher resolution results feed back on the lower resolution? Please explain further.*

We have changed this sentence to be more precise:

The domains have a spatial resolution of 27, 9 and 3 km and are 2-way nested, which allows feedbacks from the higher to the lower resolution domains.

*p.5 line 4: Does each domain has a time lag of 6 hours for initialization compared to the next bigger domain, thus an accumulated initialization of 18 hours to ERA-Interim?*

No, the time lag is the same for all three domains. We clarified this point in the manuscript by stating:

However, this relatively short spin-up period of six hours can be a drawback as the model might not be in full equilibrium. Note, the spin-up time is equal for all three domains, which means that there is no additional time lag for the nested domains.

*p.6 line 6: For which scenario(s)? How do SSTs relate to surface air temperature?*

Compared to the reference period 1961–1990, Mimura et al. (2007, chapter 16.3) projected for the fossil intensive A1 scenario a maximal warming of the Mediterranean open ocean surface air by up to 2.19 K, 3.85 K and 7.07 K for the time periods 2010–2039, 2040–2069 and 2070–2099, respectively. Additionally, Shaltout and Omstedt (2014) expected an annual warming of the Mediterranean Sea by 2.6 K and for the summer season a warming of 2.9 K by the end of the 21$^{th}$ century for the RCP8.5. Hence, the warming implied in the sensitivity experiments are in line with the spread of projected scenarios for several periods of the 21$^{th}$ century.

*p 6 line 18: Please explain in more detail. Do you interpolate both values to the same grid? Do you show the mean rate over the entire area? E-OBS doesn't have values over the ocean, however, domain D3 for which you compare precipitation includes ocean areas. How do you handle this? The cyclones don't remain in the area of interest for 5 days. How and why did you select a 5-day period?*

We are thankful for this reviewer comment since, there are certainly some shortcomings concerning the description of this part of the manuscript. We interpolated the E-OBS data to our WRF domain 3 with a bilinear interpolation technique. Since, the E-OBS data set does not include any data points over the ocean, we masked these areas in our domain 3 as well to make a fair comparison.
We decided to use 5 days, since this corresponds to the minimal lifetime of all of the five selected Vb events. It is certainly true that the precipitation over the region of interest is not fully influenced by the cyclone over this whole period. Nevertheless, since it is just a verification of the regional model and to keep it consistent with domain 3 we decided to use this 5-day period for both shown regions. Even though the cyclone is not fully influencing the region of interest for the full 5-day period, this period is useful to verify the ability of WRF to capture precipitation produced by the different features, such as cyclonic precipitation, convection and frontal systems. We included this in the paper as follows:

First, both variables are compared to observations for the entire domain 3 using E-OBS. For this the E-OBS data set is bilinearly interpolated onto the grid of the innermost domain and

the ocean grid points are masked, since the E-OBS data is land only. For the comparison, the simulated and observed mean daily precipitation rates for five Vb events are shown in Fig. 2(a).

*p.6 line 19: If you compare precipitation rates for extreme events you expect that you get different values if you use different grids. You expect lower values for a lower resolution data set as it represents the average over a larger area. (see for example Göber et al.: Could a perfect model ever satisfy a naïve forecaster? Meteorol. Appl., 15, 359 – 365, 2008)*

This is in fact a point that was lost in the argumentation and we are thankful for pointing this out. We have included the following lines:

Additionally, some of the overestimation by WRF can be attributed to the finer resolution compared to E-OBS. Hence, lower values are expected for the coarser E-OBS grid, as each grid point represents an average over a larger area compared to the WRF grid (Göber et al., 2008).

*p.6 line 27: Later you show that there is no indication of a positive bias for WRF for extreme events. With respect to biases you need to distinguish between mean and extreme precipitation.*

We have emphasised the distinction between mean and extremes in the new version by adding:

Furthermore, possible positive biases in the average precipitation of the regional model additionally increase the differences between E-OBS and WRF.

In this case, the simulated extreme daily accumulated precipitation compared to E-OBS and EURO4m-APGD, . . .

*p.7 line 29: Averaging CAPE and precipitation over the entire domain D3 obscures the signal that can be attributed to the Vb cyclones. How much do your results differ if you average over a smaller domain close to the location of the cyclone?*

We have tested several radii around the cyclone centre for the analysis, instead of averaging over the whole domain 3 (Fig. A). It turns out, that radii larger than 500 km are able to capture the full area influenced by a cyclone. Additionally, the selected radii guarantees that the noise due to the high resolution and the influence of the nearby Alps, which renders the tracking of the cyclone centre difficult, has not a strong impact. This uncertainty is aggravated in small areas, which leads to great uncertainty in the calculation of the averaged fields. Hence, radii smaller than 500 km lead to highly variable and generally unreliable results. Focussing on the average precipitation and CAPE obtained by averaging over time and space for the two radii of 500 km and 750 km, we find nearly equivalent trends as those shown in the paper for the whole domain, with few differences. CAPE monotonically increases with growing Mediterranean SSTs, while also precipitation has a clear trend towards higher precipitation with increasing SSTs. Another reason for the deviation in the trends here compared to ones shown in the paper is that, due to the high resolution of the RCM, the tracking of the cyclone is not always possible in the last hours of the cyclone existence. Hence, it is not always possible to define a cyclone centre for the time to reach the 95 % of the precipitation of the event, as shown in Fig. 4. Due to this technical but important limitation, we decided to average over the whole domain 3 which allows us to avoid the uncertainty in the definition of the cyclone centre. Furthermore, note that the definition of domain 3 is not arbitrary, but it is designed to span the area that is precisely most influenced by the Vb cyclones during their whole lifetime.

[Figure]

Figure A: Mean convective available potential energy $[10^2$ J kg$^{-1}]$ is shown in the first row for all of the ten sensitivity studies and the control simulation (black box). The second row shows the daily mean precipitation [kg m$^{-2}$ day$^{-1}$] for the same simulations. The left column depicts values for a 500 km radius around the cyclone centre and the right column for a 750 km radius.

*p.10 line 29-30: Can you confirm the results of Sodemann et al. if you look only at the 2002 case?*

Compared to the other four cases, there is actually a slightly higher sensitivity for soil moisture changes and the Atlantic SSTs for the 2002 case. Even though theses changes are not statistically significant. Hence, we do not specifically mention this in the paper.

*p.11 line 30: The statistical basis is too small for this statement.*

We have weakened this statement such that it is now only valid for our analysed cyclones:

Therefore, it seems that the five analysed summer Vb cyclones show an increasing sensitivity towards changes in the Mediterranean SSTs with decreasing maximum gradient.

*p.12 line 6: You selected the 5 strongest events, so "average" does not seem to be the appropriate term.*

The usage of the word average here seems to be confusing. Hence we try to be more precise and use the word composite, which indicates that the results are averages, what we wanted to imply and not that we analyse a regular average event. Hence we changed the sentence to:

In this study, we try to identify the main moisture source for a composite of five different high-impact Vb events.

*p.13 line 30: Some information on "maximum energy" concept for cyclones is needed here. What does it mean? How is the energy of a cyclone determined? Do you want to imply that no cyclones stronger than the selected one can appear in this region? This is a strong statement that you should check for the ERA-Interim resolution cyclones in that region.*

This is a part of the manuscript where we are being rather speculative in the interpretation of the results. We do not aim at being overly accurate, and use a rather loose definition of "cyclone energy". We have tried to emphasise the speculative nature of the argument in the reviewed version.

Our point is based on a concept of kinetic energy related to the intensity of the gradients of geopotential (and therefore the potential of the cyclone to induce geostrophic wind). Hence, we analyse the geopotential gradient of the cyclones within $1000 \times 1000$ km$^2$ around their centre, which is related to the geostrophic wind energy, i.e., the kinetic energy induced by the cyclone. Our results indicate that warmer SSTs lead to stronger kinetic energy, but this relationship is far from linear. We suggest that we are finding an upper limit, where the growth of the cyclone is somehow capped. However, we do not mean to imply that this is indeed the strongest possible cyclone in this area, but suggest that an upper end might exist. Actually, it is the Vb cyclone with the 6$^{th}$ steepest gradient in the extended summer season AMJJASO, and the 4$^{th}$ steepest when only considering MJJAS in the ERA-Interim period from 1979 to 2013. Finally, we only compare the gradients of the summer season, since the winter and summer storms have different characteristics, especially in terms of intensity (winter storms tend to be more intense than summer ones).

We included these lines in the second paragraph of section 5:

This threshold can be interpreted as energy threshold as the gradient in 850-hPa geopotential height around a cyclone is related to the wind speed (via the geostrophic approximation) and thus the kinetic energy. Therefore, our results indicate that warmer Mediterranean SSTs lead in a non-linear way to stronger kinetic energy, whereas the growth of the strongest cyclones might be capped by a possible upper energy limit.

Further we added the following lines to the summary section:

This may indicate that strong cyclones are limited in growth of kinetic energy since they might be capped by an upper bound. On the other hand there seems to be the possibility for weaker cyclones to grow in kinetic energy with increasing Mediterranean SSTs in a non-linear way. A possible reason for the limited sensitivity of strong cyclones to changes in the intensity might be, that these are more strongly steered by the large-scale atmospheric conditions, as described by Pepler et al. (2016).

*p.13 line 31: "strong cyclones are strongly steered by the atmospheric conditions" (This sentence is also included in the abstract). I don't understand what you are trying to say here and how it is a consequence of the previous statement. You need to explain this in more detail.*

Here we follow the study performed by Pepler et al. (2016), who come to similar results concerning the upper bound of energy. They also claim that the strongest cyclones are steered by large-scale dynamics rather than by the SSTs. Extra-tropical cyclones tend to be more steered by large-scale atmospheric conditions anyway, which is in contrast to the tropical cyclones, that live of the latent heating of warm SSTs. Note that we removed the sentence in the abstract as it was somehow disconnected to the sentences before.

**Technical corrections:**

*p.2 line 4: Vb cyclones are not characterized as cyclones they are cyclones.*

We changed this to: Vb events are known as cyclones that …?

*p.2 line 5-6: As they reach . . . they turn . . .*

We implemented this as suggested.

*p.4 line 30: "The increase of the Atlantic Ocean SSTs is guided by the expected changes in the Mediterranean SSTs described in Sect. 2.4.3." I don't understand this sentence. Which changes did you expect for the Mediterranean SSTs in response to the Atlantic SSTs?*

We have rephrased the sentence to:

The increase of the Atlantic Ocean SSTs in our experiment has been chosen according to the increase in SSTs in the sensitivity experiments of the Mediterranean SSTs described in Sect. 2.4.3.

*p.5 line 4: "The SSTs that are deviant compared to the control simulation are then prescribed after the vertical interpolation step of 5 meteorological data onto the domain grid." I don't understand what was done. I am surprised vertical interpolation is needed as all regionalizations use the same 50 vertical levels. I thought were adding a constant to the control SSTs. Why do you need to determine anomalies from the control? Please rephrase.*

The sentence was somewhat confusing. Hence, we rephrased it to:

The homogeneous increase in SSTs is added to the horizontally interpolated WRF grid obtained after WPS and not to the original ERA-Interim data set itself.

*p.11 line 13: Analysis and discussion of changes in cyclone (and) characteristics*

We have implemented this as suggested

*p.13 line 33: consistency not inconsistency*

We adapted this as suggested

**Detailed answer to Anonymous Referee #2**

**General comments:**

*The study aims to give a general view on the moisture sources of Vb events, but only focuses on 5 selected events from summer, which were all connected to heavy precipitation. The extreme cases are indeed interesting, but the study should underline it more that these are not typical Vb events, since even their previous study (Messmer et al., 2015) concluded that only 23 % of all Vb events are associated with extreme precipitation. Also their conclusions are valid mainly for summer, due to the event selection. This should be mentioned, since previous studies have shown that moisture sources in the Alpine region is influenced by seasonality, and for example the North Atlantic region is a more pronounced source during winter (Sodemann and Zubler, 2010). Thus the low sensitivity to the changes of North Atlantic SST might not be valid for the whole year.*

We added the point of the summer occurrence and the high-impact characteristics of the selected Vb events throughout the whole paper.

*I found the 6 hour spin-up time rather short. I would expect that the water vapour fluxes do not have enough time to adjust to the altered boundary conditions. Also Winschall et al. (2014) found that for heavy precipitation events over a slightly different domain, the time of maximum moisture uptake varies between a few hours to more than a week before the precipitation event. So with 6 hours spin up time the moisture uptake is probably already occurred, and included in the initial and boundary conditions (SST and soil moisture). So the changed boundary values and thus moisture fluxes have less effect on the cyclonic precipitation. The authors already analysed the sensitivity for the spin-up time in case of the SST experiments, but I would like to ask for more details about those analysis, and also a revision of the above-mentioned moisture uptake "problem". Also I do not see if they have investigated the spin up effect during the soil moisture experiments. I would like to see some results regarding this, because the 6 hour spin-up time also seems to be rather short for the soil experiments.*

We carried out an extensive analysis to finally opt for a 6-hour spin-up, although we only outlined it in the first version of the manuscript. We develop our findings here. Figure B might help to understand why we did not use spin-up times beyond 6hr. The second row of Fig. B shows the analysis that has been performed with nudging and with one week of spin-up time (note that nudging is necessary to guarantee for Vb cyclones given the long spin-up, but is a drawback as the cyclones are to some extent forced to follow the driving coarsely resolved data and are not fully free to independently develop, see manuscript for further details). The analysis is restricted to the four most extreme SST changes (therefore only the +/- 4 and 5 K simulations are shown). Even though there are some differences in the amplitude of the single variables, the trend is hardly distinguishable. Especially, the main variable, precipitation shows almost identical results, with only slightly more reduced precipitation for the cooling experiments. Furthermore, moisture flux from land has no influence when changing SSTs, again similar to the behaviour found with a spin-up of only 6 hours. The moisture flux from the ocean is generally smaller with a longer spin-up time, but nevertheless, the non-linear behaviour with changing SSTs is a robust feature consistent with shorter spin-up time. Precipitable water is as well indistinguishable to the one observed with shorter spin-up times. CAPE is much stronger in the longer simulations; as the simulation has a longer time available to create atmospheric

[Figure]

Figure B: The two rows show the mean over 5 Vb events for the Mediterranean SST experiments with bars depicting a decrease in SSTs of 5 K on the left to an increase in SSTs of 5 K on the right. The first row shows the experiment with 6hrs spin-up time and no nudging, while the second row depicts the experiment with nudging and 1 week of spin-up time. The five columns show the daily mean precipitation, upward moisture flux over land and over the ocean, and mean convective available potential energy (CAPE) for D3 from the left to the right. The units for the y-axis are given in the header of each column, whereas the x-axis denotes the performed sensitivity studies.

instabilities, with a longer heating time. However, it can be argued that such an extended heating, is indeed not physically consistent with respect to the control run, a fact that we point out to in our summary. This is an important reason for choosing the shorter spin-up times. All in all, the findings we discuss in this study are insensitive to the length of spin-up time beyond the 6 hours we finally selected. To keep the study as condensed as possible we opted for minimising the size of this discussion in the manuscript.

Nevertheless, as already stated in the first answer to the reviewer, we have solid scientific reasons not to use longer spin-up times for the soil moisture experiments. In this case, we want to analyse whether reduced soil moisture, as the one that might happen in a future climate, does influence the precipitation during Vb events. To get a strong-enough signal that allows exploring the sensitivity of the precipitation to this factor, we run two highly idealized extreme experiments, where we removed and saturated the soil moisture completely in all four soil model layers. These types of experiments are fundamentally different in nature to the SST experiments. In the SST experiments, we change a given boundary condition, whereas in the case of soil, the variables are simulated online together with atmosphere model, and therefore the experiment consists of changing an initial condition within the soil model. In this case increasing the spin-up time is undesirable, as this period would be used by the model to refill the soil moisture volume until the equilibrium is recovered. This would render the setup of the experiment useless to analyse the sensitivity of the simulation to the original perturbation. This is especially true for the first model soil layer, which is the most weather-relevant layer and the one with shortest response time. We acknowledge however that this is a subtle yet important point that we did not discuss in enough detail, so we have emphasised it including the following lines in section 2.4.1:

For this reason, we did not use longer spin-up times than 6 hours, since the model would use

the longer spin-up period to refill the soil moisture volume until the equilibrium is recovered. This is especially true for the first model soil layer, which is the most weather-relevant layer and the one with shortest response time. Furthermore, note that in the SST experiments, we change a given boundary condition, whereas in the case of soil, the variables are simulated together with the atmosphere model, and therefore the soil experiments are experiments where the initial conditions are changed.

**Specific comments:**

*Abstract: The soil moisture experiments are mentioned, but no results are included, also the North Atlantic experiments are not mentioned.*

We included a line in the abstract stating that the analysed Vb events are rather insensitive to changes in the Atlantic SSTs and soil moisture.

*Page 2 Line 8: Besides the Elbe, it would be nice to mention, which other large rivers are especially affected by the precipitation of the Vb cyclones.*

We have also included the other large rivers that can be affected such as the Danube and the Rhine.

*Page 2 Line 30-32: Please reformulate this sentence so, that is not so strong, it should show that the variability of moisture sources are still high besides the seasonality.*

We have tried to weaken the statement such that it reads like this:

Nevertheless, Winschall et al. (2014) further supported the fact that the Mediterranean Sea is not the only moisture source during various heavy precipitation events in Central Europe. They found that additional moisture sources with high event-to-event variability are needed to trigger such events. These moisture sources include beside others the evaporation from European landmasses especially in summer or evaporation from the North Atlantic Ocean in winter.

*Page 3 Line 8: Please write "extreme Vb events" instead of just "Vb events".*

We have even included the point of the summer season here by stating:

extreme summer Vb events.

*Part 2.1: Can you include here some information about the SST field? If not the ERA Interim SST fields are used for the sensitivity experiments, then please write something about the details of the SST boundary fields at the model setup part.*

We believe that actually the misunderstanding occurs in section 2.4.3. Hence, we have added some lines to be more precise on the point with the SST adaption in the sensitivity studies:

The ERA-Interim SST field is used to calculate the horizontally interpolated SST field for the input file used by WRF. The homogeneous increase in SSTs is then added to this input file on the WRF grid and not to the original ERA-Interim data set itself. This is done to avoid any inconsistencies in the increased Mediterranean SSTs at grid points close to the coast lines, related to differences in the land-sea mask of the ERA-Interim and WRF domain.

*Part 2.2: Please mention, which variables do you use from E-OBS. Also I found the terminology for precipitation rate a bit misleading. As I understood precipitation rate here means daily accumulated precipitation, and accumulated precipitation means multi-day sums of daily precipitation.*

We added the following sentence to section 2.2:

For our analysis we will only use the daily accumulated precipitation.

Furthermore, we have changed the terminology in sect. 3.1 according to the reviewer's suggestions.

*Part 2.3: It might be useful to shortly introduce the synoptic situations regarding the selected events, e.g. what was different and what was the same for the 5 cyclones.*

We have included some sentences concerning the synoptic situation of the five Vb events in general. Since this is not a case study and because we look at these events mostly as composites, we think that a description of each single event is beyond the scope of this study. Also, the Vb cyclone itself is already some sort of a synoptic situation, which is actually already quite specific. Nevertheless, we have included the following lines to be more precise on the large-scale situation of the analysed five Vb events:

All of the five events are initialised by a cold air outbreak located northeast of the Alps. As this trough moves westwards lee-cyclogenesis is induced at the southeastern flanks of the Alps and hence in the region of the Gulf of Genoa. From this starting point all of the five analysed Vb cyclones move along the Vb track described by Van Bebber (1891), showing some individual behaviour along the path of course.

*Part 2.4: Can you include more information about, how the atmosphere interacts in the model with the SST boundary conditions and with the soil (e.g. frequency, fluxes).*

The moisture is transported to the atmosphere through the surface latent heat flux, which is calculated by the surface layer parameterisation. The latent heat flux and also the sensible heat flux are exchanged every time step. Since these are standard technical details, we have decided not to include more information on this in the paper.

*Page 4 Line 21: Please explain in more detail, what does 2-way basis mean. Is it 2 way nesting?*

We have changed this sentence to be more precise:

The domains have a spatial resolution of 27, 9 and 3 km and are 2-way nested, which allows feedback to happen from the higher to the lower resolution domains.

*Page 5 Line 9: Please state clearer that the spectral nudging is done for these extra spin-up sensitivity experiments, otherwise it is a bit confusing after stating that nudging techniques are avoided (Page 4 Line 30).*

We made this clearer by adding the following sentence:

Note, that nudging has only been applied to this one week spin-up setup.

*Page 6 Line 1: Mentioning eleven simulations is misleading, since the control is not considered as a Mediterranean SST sensitivity experiment. So please change to "10 sensitivity and one*

*control" or "10 sensitivity simulations".*

We changed this to "ten sensitivity simulations plus a control simulation"

*Page 6 Line 7: Can you find a projection for SST instead of surface air?*

We included the following study to our paper and hence, added the following sentence:

Additionally, Shaltout and Omstedt (2014) expected an annual warming of the Mediterranean Sea by 2.6 K and for the summer season a warming of 2.9 K by the end of the 21$^{th}$ century for the RCP8.5.

*Page 6 Line 23: Higher precipitation rates can also be due to the higher resolution of the simulated data.*

This is in fact a point that was lost in the argumentation and we are thankful for pointing this out. We have included the following lines:

Additionally, some of the overestimation by WRF can be attributed to the finer resolution compared to E-OBS. Hence, lower values are expected for the coarser E-OBS grid, as each grid point represents an average over a larger area compared to the WRF grid (Göber et al., 2008).

*Page 7 Line 20: Please denote somehow on the tracks in Fig. 3, which are the intensification and decaying phases of the cyclones.*

We have included a diamond at the point when the cyclone has reached the deepest gradient. We have done this only for the control simulation (green line) and the ERA-Interim track (black line). We believe that the plot is already quite busy with all the tracks of the sensitivity studies, such that putting diamonds for all of the sensitivity experiments would strongly decrease the readability of Figure 3. Still the timing of the deepest gradient is similar in the sensitivity experiments.

*Page 7 Line 30. The moisture uptake from land and ocean by the cyclones happened probably before the precipitation. I think time steps before the precipitation can also give information about the moisture exchange.*

The analysed period includes the whole simulation, and thus, at the point when the cyclone develops, until 95 % of the entire event accumulated precipitation has fallen. Obviously this period also includes time steps (one day or even more) before the actual heavy precipitation events starts, in which the model has time to uptake moisture that can be then used to increase the precipitation output.

*Page 8 Line 17: Why are the ocean-land winds slightly reduced?*

We have decided to remove this sentence, since it seems to be rather confusing than helpful and because it is about changes that are not even significant. Thus, we have replaced these sentences with the following:

There is a slight reduction (increase) in the mean upward moisture flux over the Mediterranean Sea. These changes are not significant and hence, their changes are not analysed in more detail here.

*Page 9 Line 8: Note that the North Atlantic is shown to be more important during winter precipitation events. So there is maybe a lack of sensitivity because these were summer events.*

This is a good point and we are really thankful for this reviewer comment. Since this is part of the interpretation of the results, we have decided to add the following sentences in Discussion part rather than in the results part:

The insensitivity of the analysed Vb events to Atlantic SST changes, might also be due to the fact that they are all observed during summer. It seems that the Atlantic Ocean might steer the atmospheric moisture stronger in winter (Sodemann and Zubler, 2010).

*Page 10 Line 5-16: It would help the understanding if an extra domain, a costal domain, would be introduced, and the results would be visualised in a way similar to Fig. 4k.*

We have plotted the values similar to Fig. 4k, but only considering a box over the Adriatic

[Figure]

Figure C: Daily mean precipitation for the different sensitivity studies with the Mediterranean SSTs are shown for the Adriatic coast only.

coast (Fig. C). The precipitation over the Adriatic coast is strongly increasing with increasing Mediterranean SSTs. Nevertheless, we believe that we do not really gain much from this, since the essence of this behaviour is already depicted in Fig. 5f and 5g, containing additionally the information on the spatial pattern. For this reason we will not increase the number of figures in the present study.

*Page 10 Line 20: 24 % in which direction, and where?*

We are happy that the reviewer pointed out this shortcoming. We have extended the sentence to:

This is because an increase of 5 K in the Mediterranean SSTs leads to a rise in precipitation of up to 24 % over Central Europe.

*Figure 4c. Moisture flux over ocean, is misleading, since almost all points are from the Mediterranean Sea. It might be clearer if the few North Atlantic Ocean points would be excluded, and the moisture flux would only refer to the Mediterranean Sea.*

We have implemented the suggestion and have added a sentence in the beginning of section 4 to clarify the origin of the presented values:

Since most of the ocean grid points are located over the Mediterranean Sea and only few over the Atlantic (see domain 3 in Fig. 1) these few grid points have been masked to obtain only the

moisture flux over the Mediterranean Sea.

*Figure 6. Please mention the resolution of the different data.*

We have implemented this as suggested.

**References**

[revised manuscript text omitted]

---

## Referee Report (RR1)

Review of the study "Sensitivity Experiments on the Response of Vb Cyclones to Ocean Temperature and Soil Moisture Changes"
from
Martina Messmer, Juan José Gómez-Navarro, and Christoph C. Raible

The manuscript has improved through the changes, and I see that the Authors considered my comments. Although in one aspect, regarding the soil moisture sensitivity experiment, I still want to ask the Authors to reconsider their simulations, or if different sensitivity experiments are not possible reconsider some of their statements.

My main concern regarding the soil moisture experiment is that, by starting the model with 6 hours spin up time, the model is initialised with atmospheric moisture, which is partially evaporated from the land areas during the previous days. In other words in this kind of sensitivity experiment I do not think that one can investigate the full effect of the soil, as moisture source for these cyclones, since its moisture is partially included through the atmospheric initial and boundary conditions. Maybe this is also the reason, why this experiment did not show such significant changes as the Mediterranean SST experiment. In my opinion, changing the soil relatively close to the cyclones' initialisation, shows mainly the effect of moisture recycling during the cyclones' lifetime.

I would recommend for example to try and start the model (in case of the soil moisture experiments) from a previously simulated dry soil experiment, where the soil moisture have been decreased not just in the initial conditions, but during the integration of the model, and thus the atmosphere had time to adapt to the altered dryer or more moist conditions. This could be achieved by changing the parametrisation of soil moisture or at the calculation of surface fluxes. Of course this can cause other problems with the cyclones' initialisations.

An other solution could be to make the soil moisture experiment more similar and comparable to the SST experiments, by drying/irrigating the soil in every time step, in a similar manner as the SST field is kept altered during the whole simulation.

If such an experiment is not possible, then I would like to as the Authors to change some of the sentences in the text:

1. Abstract Line 5-6: I think that the statement that the Vb events are rather insensitive to the soil moisture changes, can not be concluded from these results, please reformulate it, because in this form I think it is too strong and possibly misleading.

2. Section 4.1, Page 9 Line 15-17: It might be worth mentioning that the dry soil experiment shows significant changes (Fig. 4a and b).

3. Section 4.4, Page 11 Line 26-27: Same as in the Abstract.

4. Section 4.4, Page 11 Line 30-32: The soil experiment is called unrealistic, but on the other hand the +5K SST experiment is also a rather unrealistic scenario. So I am not convinced that justifying the conclusions by calling them unrealistic is correct here, especially because the other sensitivity experiments are similarly unrealistic.

5. Section 4.4, Page 12 Line 6: The reason for marginal contributions from the soil can be due to that, the moisture originating from the soil was already initialised through the atmospheric moisture. See my main comment.

6. Section 6 Page 14 Line 13-15: See the 4th comment.

7. Section 6 Page 14 Line 23: Both experiments are extreme. Here, you consider the results from the soil moisture experiment equally important to the +5K Mediterranean SST results, but the rest of the paper is considering the Mediterranean much more important. I feel that the conclusions are biased towards the Mediterranean in a way, which is not in accordance with the results. Also since climate projections not only predict the increase of Mediterranean SST but also dryer conditions (so dryer soil) in this region, the effect of soil moisture can not be "ignored" in the Mediterranean region.

---

## Editor Decision (ED1)

**ESD-2016-67: Editor Decision (ESD)**

February 8th, 2017

Dear Authors,

Thank your very much for your comments and developments in response to the reviewer reports, and for taking into consideration the preliminary editorial recommendations made prior to the Interactive Discussion stage.

After careful consideration of the manuscript, the reviewer reports and the author comments, an editorial decision can now be formulated.

Overall, the scientific questions addressed in the manuscript are relevant to the interdisciplinary scope of Earth System Dynamics, and significant efforts have been made by the authors in addressing such questions in their submitted work.

Albeit its good merits, the manuscript would benefit from further clarification and revision in order to address the pertinent concerns raised during the peer-review process.

In this regard, the diligences conducted in response to the reviewers are already an encouraging step forward. These should thus proceed towards addressing the raised concerns and further strengthening the arguments of the paper.

Key aspects debated at the Interactive Discussion are also encouraged to be incorporated in the revised manuscript. In fact, some questions raised by the reviewers could also arise among the broader readership, and the author responses provide good grounds to further elaborate and substantiate their claims in the revised manuscript.

In order to provide enough time for a careful revision further harnessing the valuable potential of this manuscript, my decision entails "major revisions".

I will be looking forward to the revised manuscript.

With very best wishes,

Rui Perdigão
(ESD Editor)

---

## Author Response (AR2)

**Detailed answer to Fanni Dora Kelemen**

**General comments:**

*My main concern regarding the soil moisture experiment is that, by starting the model with 6 hours spin up time, the model is initialised with atmospheric moisture, which is partially evaporated from the land areas during the previous days. In other words in this kind of sensitivity experiment I do not think that one can investigate the full effect of the soil, as moisture source for these cyclones, since its moisture is partially included through the atmospheric initial and boundary conditions. Maybe this is also the reason, why this experiment did not show such significant changes as the Mediterranean SST experiment. In my opinion, changing the soil relatively close to the cyclones' initialisation, shows mainly the effect of moisture recycling during the cyclones' lifetime.*

Thanks for the clarification. The caveats pointed out by the reviewer are now clearer to us. Hence, we have tried to improve this in the current version of the manuscript as detailed below.

*I would recommend for example to try and start the model (in case of the soil moisture experiments) from a previously simulated dry soil experiment, where the soil moisture have been decreased not just in the initial conditions, but during the integration of the model, and thus the atmosphere had time to adapt to the altered dryer or more moist conditions. This could be achieved by changing the parametrisation of soil moisture or at the calculation of surface fluxes. Of course this can cause other problems with the cyclones' initialisations.*

To address this point, we have redesigned the setup of the soil moisture sensitivity experiment, which involved hard-coded changes in the source code of the model. For this experiment we perform a pre-simulation to the actual simulation of the Vb events, which is used just to produce atmospheric humidity variables more consistent with the soil moisture changes. The pre-simulation uses however nudging to guarantee a Vb cyclone development under the newly obtained moisture level. This pre-simulation starts five days prior to the actual simulation of the Vb events and lasts for five days. These five days allow the atmosphere to adjust to the new soil moisture conditions of the different experiments. To make the sensitivity experiment even more clear, during the pre-simulation the soil model is modified in a way such that it does not allow accumulating soil moisture content at all in any of the four available soil layers for the desaturation experiments. Similarly, for the complete saturation the soil moisture content is artificially kept at the maximal possible soil water content that is allowed for the respective soil types. We then take the water vapour variables (water vapour in the atmosphere and at 2 meters) of the pre-simulation's output and use them as initial conditions for the actual Vb simulation, in which the rest of the variables are taken from the driving reanalysis as in a regular simulation. We hope this experimental setup overcomes the reviewers concerns about the moisture content in the atmosphere that is originating from the soil. We have included a new paragraph in the text that thoughtfully explains this experimental setup:

"To test the sensitivity of Vb events to soil moisture, three different experiments are carried out. They enclose a complete desaturation to the minimum possible soil moisture

content of 2 %, fixing a soil moisture content typical of southern Spain homogeneously across the whole domain (i.e. 17.5 %, which corresponds to the average value of the analysed Vb events in the region of Ciudad Real, being this one of the driest regions in the Iberian Peninsula), and a complete saturation of the soil moisture content. The second setup is a very unrealistic, yet physically plausible scenario, and therefore can be regarded as a more realistic version of the complete desaturation of soil. Note that all of the three performed experiments are rather unrealistic and highly idealised, and are aimed at exploring physical mechanisms, rather than obtaining accurate climate change projections. A complete desaturation of all the soil moisture throughout whole Europe is probably the most unrealistic one, since it comprises soil water contents over whole Europe that do not even occur in the Saharan Desert. It is possible that Central Europe could see a general drying to a more Mediterranean climate (Seneviratne et al., 2010), but nevertheless it is quite a strong reduction in soil water content for most of the land area covered by domain 1. In this sense, the second experiment is a slightly less unrealistic version, although still very unlikely given current climate change projections in the Mediterranean Sea. The projections for Central Europe indicate a robust 5-15 % reduction in soil moisture for the end of the century, with a tendency for wetter soils in the northern parts of Europe (Seneviratne et al., 2010). This projected reduction is still higher than the southern Spain experiments, as a reduction of 15 % of the Central European soil moisture would result in our cases in 25 % soil water content. Similarly, the full saturation experiment is also rather unrealistic even in a possible moistening scenario of Europe.

Since the evaporation from soil moisture can influence the moisture content in the atmosphere before the actual Vb event takes place, we have carefully designed the initialization of these simulations. For all the three experiments, as well as the control simulation, the WRF model is started five days before the actual Vb event is initialized and terminated after these five days. During this pre-simulation, we use the same spectral nudging as described for the SST test simulations, and the soil moisture is constantly overruled to impose a fixed value of soil moisture according to each of the three sensitivity experiments in all four model layers of the Noah model. The atmospheric water vapour content after these five days of the pre-simulation is then used to overwrite the water vapour present in the initial conditions taken from the driving dataset and used in the actual Vb simulation. The actual Vb event simulations are started at the same time as the SST experiments in order to obtain similar cyclone tracks throughout the different types of experiments and therefore minimise side effects arising from changes in the Vb dynamics.

Across all soil moisture sensitivity experiments, the initial conditions for soil moisture in the actual simulation are set to the corresponding value according to each the three families described above. In this regard, it is important to note that just the initial conditions are set, i.e. the model is free to adjust the soil moisture afterwards due to e.g., precipitation and evaporation processes. For this reason, we did not use spin-up times longer than 6 hours, since otherwise the model would use the longer spin-up period to refill the soil moisture volume until the equilibrium is recovered. Further, such short spin-up precludes obtaining a realistic initial condition of the water atmosphere content in equilibrium with the perturbed soil, which is the reason for running the pre-simulations described above. The care taken in the initialization of the soil experiments

pertains especially the first model soil layer, which is the most weather-relevant layer and the one with shortest response time. It is important to remark that unlike in the SST experiments, where we change a given boundary condition, in the case of soil the variables are simulated together with the atmosphere model, and therefore the soil experiments shall be regarded as perturbation in the initial conditions. To change the soil moisture content for the actual Vb event simulation, the original ERA-Interim initial file is modified and the land values are set to either 0, 0.175 or 0.5 $m^3$ $m^{-3}$. The latter value is selected, because the soil moisture content of all soil types listed in the WRF model is always lower than 0.5 $m^3$ $m^{-3}$"

Furthermore, we have made small changes to adjust the results throughout the text, which are highlighted in the difference pdf-file.

*Another solution could be to make the soil moisture experiment more similar and comparable to the SST experiments, by drying/irrigating the soil in every time step, in a similar manner as the SST field is kept altered during the whole simulation.*

We did not consider this type of experiment as we wanted to allow for moisture recycling during the Vb events and because this is a mechanism we did not want to prescribe.

*If such an experiment is not possible, then I would like to as the Authors to change some of the sentences in the text:*

As we conducted a whole new set of experiments to overcome the original issues pointed out in the review, it is not necessary to accomplish the changes suggested as an alternative to performing new experiments.

*The soil experiment is called unrealistic, but on the other hand the +5K SST experiment is also a rather unrealistic scenario.*

We would like to shortly comment on this statement. It is true that a 5-degree warming of the Mediterranean Sea for end of century seems unrealistic under current climate change scenarios. Nevertheless, this scenario seems to be possible if we look at longer-term climate variability such as by the end of the 22nd century. In contrast, the complete desaturation experiment considers soil moisture values, which are even lower than the driest points that can be observed in the Saharan desert. It seems to be extremely unlikely, and even physically impossible, that such conditions can be reached at any time even in the farer future, since it would render Central Europe as an unsuitable place for most living species. In order to generate a somewhat more comparable experiment to the 5 degree warming of the Mediterranean Sea, we decided to include an additional soil moisture sensitivity experiment, which is now included in the analysis. For it, we used a southern Spain soil moisture condition. For the pre-simulation of this simulation we kept the soil water volume constantly at 17.5 %. The results indicate that even this rather unrealistic and strong drying of the whole European continent, does not strongly and significantly affect the precipitation amounts during the analysed Vb events.

[revised manuscript text omitted]

Dear Mr Perdigão,

Thank you very much for the careful review of our revisions on "Sensitivity Experiments on the Response of Vb Cyclones to Ocean Temperature and Soil Moisture Changes". We are thankful for the constructive comments on this manuscript.

*Page(P.) 6, line(l.) 30: Regarding RCP 8.5, it would be helpful for the less-informed readership to add a small parenthesis noting that this is a worst-case scenario emerging from particularly pessimistic assumptions and associated parameter setup in climate projections. Consistency of extreme tests (e.g. massive SST increases) with this scenario is logical and worth noting given the extreme nature of the events focused in this study.*

We have included this point in the manuscript as suggested.

*P.10, l.27: "a non-linear relationship further discussed below": Even though the non-linearity in the precipitation response to the thermal marine forcing is mentioned several times in the paper, it is essentially described as "non-linear way" without specific detail about the underlying non-linear functional relationship. Addressing this would further strengthen the discussion.*

We didn't originally aim at providing a comprehensive description of this phenomenon and its underlying reasons, as we consider it a rather off topic that might obscure the main message of the manuscript. Still, we mention the non-linear behaviour to emphasise that the functional relationship between both variables is not simply proportional, as could be expected under a very shallow analysis.

In the manuscript we actually give a possible reason for the non-linear relationship between the precipitation response and the thermal marine forcing. Hence, we argue that this non-linear increase in precipitation is related to the increase in convective available potential energy (CAPE) as stated in section 4.3.:

As indicated above, the Mediterranean SST sensitivity experiments exhibit a non-linear increase in precipitation amounts in domain 3 with increasing SSTs (Fig. 4(k)). This can be due to two different mechanisms. One is the increased moisture flux induced by increased SSTs. This increased moisture flux leads to a mostly linear increase in the average atmospheric moisture, as demonstrated by the amount of precipitable water in Fig. 4(n). Nevertheless, the nonlinear behaviour observed in the average precipitation is driven by an increase in atmospheric instability, i.e., CAPE. Hence, an increase in atmospheric water vapour goes along with an increase in latent heat and leads to additional convection, which is capable of removing an even larger portion of water than expected from the single increase in atmospheric moisture.

*P.11, l.31: "It seems that the Atlantic Ocean might steer the atmospheric moisture stronger in winter Sodemann and Zubler (2010)." → Consider rephrasing along this line: "This is consistent with the argument that the North Atlantic might influence the atmospheric moisture more strongly in winter (Sodemann and Zubler, 2010)".*

We have rephrased the sentence as suggested by the Editor.

*Detecting low sensitivity in the analysed Vb events to Atlantic SST changes in summer does not necessarily imply that the Atlantic has low influence on the events taking place during that season. Undetected Atlantic influences may be manifested through mechanisms elusive to the analysis, such as ocean-atmospheric fluxes conditioning the hemispheric-scale mid-latitude atmospheric dynamics where the regionally emerging Vb cyclones are nested.*

We fully agree with this statement and we think that we are clear that our analysis focuses on the direct influence of SSTs on Vb cyclones 6 h before they are detected. Clearly the Atlantic plays a role in preparing the atmosphere to generate Vb cyclones, however this was not the focus of this study as the focus is on the precipitation impact of these events. After rereading the manuscript, we tried to make this clear, e.g. we changed the title to 'Sensitivity Experiments on the Response of Vb Cyclones to **Sea Surface** Temperature and Soil Moisture Changes'. Further we applied the following to the discussion (section 4.4):

'Still this does not mean that the Atlantic has no influence on the Vb cyclones throughout a season as seasonal SST change might change the atmospheric circulation stimulating the generation of Vb cyclone. Nevertheless, such responses are not possible to asses with the experimental design selected and are thus beyond the scope of the study.'

We are very grateful for the comments for the technical notes, which we have included in the revised manuscript.

Best wishes,

Martina Messmer

---

## Editor Decision (ED2)

**ESD-2016-67R: Editor Decision**

April 10th, 2017

Dear Authors,

Thank you very much for all your efforts and diligence in carefully preparing an improved manuscript, taking into consideration the concerns raised in the peer-review process, and providing thoughtful and comprehensive responses to the evaluation reports.

The referee that kindly re-evaluated the manuscript acknowledged the improvements and was supportive of publication conditional to addressing the raised concerns mostly related to the soil moisture sensitivity experiment.

Further to the comprehensive referee report, I would note the following aspects for consideration in a *Minor Revision*:

- Page(P.) 6, line(l.) 30: Regarding RCP 8.5, it would be helpful for the less-informed readership to add a small parenthesis noting that this is a worst-case scenario emerging from particularly pessimistic assumptions and associated parameter setup in climate projections. Consistency of extreme tests (e.g. massive SST increases) with this scenario is logical and worth noting given the extreme nature of the events focused in this study.

- P.10, l.27: "a non-linear relationship further discussed below": Even though the non-linearity in the precipitation response to the thermal marine forcing is mentioned several times in the paper, it is essentially described as "non-linear way" without specific detail about the underlying non-linear functional relationship. Addressing this would further strengthen the discussion.

- P.11, l.31: "It seems that the Atlantic Ocean might steer the atmospheric moisture stronger in winter Sodemann and Zubler (2010)." → Consider rephrasing along this line: "This is consistent with the argument that the North Atlantic might influence the atmospheric moisture more strongly in winter (Sodemann and Zubler, 2010)".

- Detecting low sensitivity in the analysed Vb events to Atlantic SST changes in summer does not necessarily imply that the Atlantic has low influence on the events taking place during that season. Undetected Atlantic influences may be manifested through mechanisms elusive to the analysis, such as ocean-atmospheric fluxes conditioning the hemispheric-scale mid-latitude atmospheric dynamics where the regionally emerging Vb cyclones are nested.

In addition to these, I leave the following short technical notes:

- P.2, l.16: "Furthermore, the large-scale dynamics seem to determine, if a Vb cyclones delivers high precipitation or not" → consider removing the second comma and slightly rephrasing as: "Furthermore, the large-scale dynamics seem to determine whether a Vb cyclone delivers high precipitation or not"

- P.5, l.17: "Note, the spin-up" → consider rephrasing as: "Note that the spin-up"

- P.5, l.25: "Note, that" → "Note that" (no comma)

- P.5, l.34: "longer spin-up times than 6 hours" → "spin-up times longer than 6 hours"

- P.6, l.1-3: Unnecessary commas can be removed in: "SST experiments[,] we"; "soil[,] the variables".

- P.12, l.8: "our analysis are" → "our analyses are" (plural subject concordant with verb)

- P.14, l.29-30: "Hence, a non-linear behaviour in precipitation is found, and can be attributed to" → "Hence, a non-linear behaviour is found in the precipitation sensitivities, attributable to".

I look forward to the revised manuscript.

With very best wishes,

Rui Perdigão
(ESD Editor)